# REPRESENTATION INVARIANCE AND ALLOCATION: WHEN SUBGROUP BALANCE MATTERS

## ABSTRACT

Unequal representation of demographic groups in training data poses challenges to model generalisation across populations. Standard practice assumes that balancing subgroup representation optimises performance. However, recent empirical results contradict this assumption: in some cases, imbalanced data distributions actually improve subgroup performance, while in others, subgroup performance remains unaffected by the absence of an entire subgroup during training. We conduct a systematic study of subgroup allocation across four vision and language models, varying training data composition to characterise the sensitivity of subgroup performance to data balance. We propose the *latent separation hypothesis*, which states that a partially fine-tuned model's dependence on subgroup representation is determined by the degree of separation between subgroups in the latent space of the pre-trained model. We formalise this hypothesis, provide theoretical analysis, and validate it empirically. Finally, we present a practical application to foundation model fine-tuning, demonstrating that quantitative analysis of latent subgroup separation can inform data collection and balancing decisions.

## 1    INTRODUCTION

There is a wide consensus in machine learning that model performance improves monotonically with increasing training data (Rosenfeld et al., 2020; Kaplan et al., 2020). This principle, formalised through dataset scaling laws, has guided much of the recent progress in model training. However, real-world data rarely satisfies the assumption of being independent and identically distributed (i.i.d.) (Arjovsky et al., 2020; Wang et al., 2023). Instead, datasets are composed of clusters of correlated samples, corresponding to subgroups or domains. In the medical domain, clusters may correspond to demographic categories, while in image datasets they may reflect camera types, and in multilingual corpora they may represent language varieties.

In such cases, the question becomes more nuanced: how does model performance on a particular subgroup scale as its representation in the training data increases? While intuition suggests that a higher proportion of subgroup-specific data should directly improve performance on that subgroup, recent studies have revealed surprising counterexamples (Rolf et al., 2021; Weng et al., 2023; Čevora et al., 2025), where increasing subgroup allocation had little or even no effect. This challenges the widely held view that dataset rebalancing is always a reliable solution (Idrissi et al., 2022).

Therefore, understanding the relationship between subgroup allocation and subgroup performance remains an important open question. When concerned about model fairness, practitioners must decide whether to conduct certain interventions, like collecting balanced data across demographic groups, or resampling or augmenting their dataset, potentially at the cost of reduced overall performance (Raji & Buolamwini, 2019; Idrissi et al., 2022). In domain generalisation, practitioners must weigh whether fine-tuning on a smaller set of domain-specific data will yield better deployment performance than fine-tuning on a larger set of general data (Hulkund et al., 2025). More broadly, given the cost of data collection and annotation, knowing when subgroup representation matters can guide whether to prioritise general high-quality data or group-specific data.

In this work, we aim to understand how the allocation of data across subgroups affects subgroup performance at inference time, given a fixed training budget. Through extensive experiments in vision and language tasks, we find that subgroup sensitivity to allocation varies dramatically across

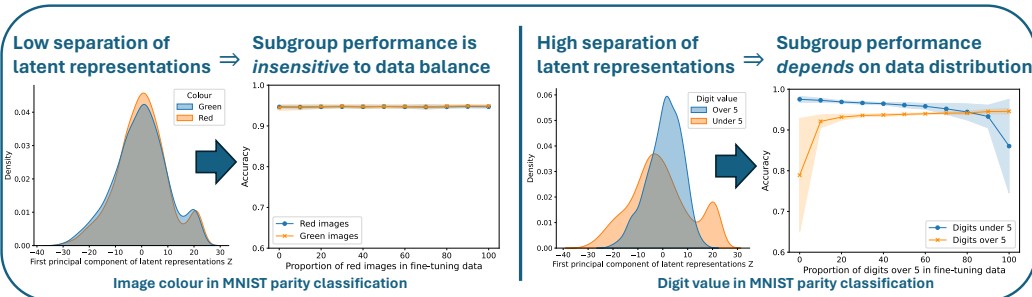

Figure 1: **Model sensitivity to data balance depends on latent separation of subgroups.** Left plots show PCA projections of latent representations of MNIST parity classifiers. Right plots show subgroup accuracy as training data allocation changes.

datasets, models, and attributes. We probe *why* these differences arise and put forward a novel hypothesis: the degree to which redistributing subgroup data improves subgroup performance is determined by how strongly the set of subgroups are separated in the pre-trained model's latent representations. We provide both theoretical justification and empirical evidence of this hypothesis.

**Our contributions are:**

§4 We demonstrate that widely held explanations for sensitivity to subgroup allocation fail to match empirical behaviour.

§5 We derive a theoretical upper bound on sensitivity to subgroup allocation based on subgroup separation in the pre-trained model's latent space.

§6 We show empirically that sensitivity to subgroup allocation is significantly correlated to the distance between two subgroups in the pre-trained model's latent representations.

§6.5 We show how our findings can guide dataset selection decisions to improve fairness in a practical case-study fine-tuning a vision-language foundation model.

## 2 RELATED WORK

**Dataset scaling is not straightforward when the train and deployment settings are not i.i.d.** We are broadly interested in the relationship between model training data and model performance. Research in this area has investigated different aspects of data (e.g., size, composition, or individual points) and different performance metrics (e.g., overall loss, fairness, or domain-specific accuracy) Hashimoto (2021). While dataset scaling laws have shown that performance improvements follow predictable power law trends (Rosenfeld et al., 2020; Kaplan et al., 2020), this relationship becomes more complex when the training and test data are drawn from different distributions. In such cases more data is not necessarily better. For instance, Hulkund et al. (2025) and Shen et al. (2024) both show that when optimising for a specific deployment setting, a subset of the data can yield better performance than the full dataset. Similarly, Diaz & Madaio (2025) argue that scaled training data can have a negative impact depending on the evaluation metrics and subpopulations considered.

**Subgroup data scaling through the lens of fairness** This problem has also been studied indirectly in the field of fairness, where data are grouped into subgroups (e.g., based on demographic attributes), and one investigates how training data composition (i.e., number/proportion of samples from certain subgroups) affects fairness (i.e., some metric based on model performance on certain subgroups). The prevailing assumption is again that more subgroup-specific training data leads to improved performance on that subgroup (Raji & Buolamwini, 2019; Chen et al., 2018). When more data cannot be collected, the standard intervention is to rebalance the model training data by under- or over-sampling samples from certain subgroups (Idrissi et al., 2022). Many works show that this simple method can yield remarkable fairness improvements both when training from scratch (Idrissi et al., 2022) and when fine-tuning (potentially biased) pre-trained models (Kirichenko et al., 2023; Wang & Russakovsky, 2023; Alabdulmohsin et al., 2024).

**Inconsistencies in subgroup balancing results** However, a growing body of work argues that balancing data does not necessarily improve fairness, and that it can even be detrimental. Schrouff et al. (2024) use a causal framework to show conditions under which data balancing will *not* improve model fairness. Similarly, Schwartz & Stanovsky (2022), Roh et al. (2021), Qiao et al. (2025), and Claucich et al. (2025) show that fairness is not necessarily maximized at uniform group ratios, with the latter arguing that this is due to unequal task difficulty across groups. Weng et al. (2023) and Čevora et al. (2025) even show cases where a model's performance on female medical images remains constant (and sometimes decreases) as the proportion of female images in the training set increases. Loss-reweighting based approaches such as group distributionally robust optimisation (Sagawa et al., 2020) are based on a related principle: rather than balancing group size, they reweight groups with higher losses. However, loss-based methods also exhibit failure modes (Zong et al., 2023), and it remains unclear whether up-weighting data from poor-performing groups necessarily improves performance on those groups. Together, these studies reflect an emerging trend in fairness research, that different subgroups have distinct properties and causes of under-performance, and therefore respond differently to interventions like data balancing or loss re-weighting (Wang, 2025; Alloula et al., 2025; Jones et al., 2024; Yang et al., 2023). However, without a causal framework (which is difficult to apply in practice) (Jones et al., 2024; Schrouff et al., 2024) or direct experimentation, it is difficult to determine *a priori* whether balancing will improve fairness.

**Impact of subgroup allocation on subgroup performance** Our work differs from fairness-oriented studies in that we address a more fundamental question: how does subgroup allocation affect subgroup performance? We argue that understanding this is prerequisite for tackling fairness concerns and implementing any bias mitigation methods. Despite its importance, this question has received little direct attention, and as discussed above, it is not clear whether redistributing data from under-represented or poorly performing groups reliably improves performance for those groups. Rolf et al. (2021) take a first step by fitting a per-group power-law scaling model describing the impact of subgroup and total training data size on subgroup performance. Similarly to the fairness papers, they show that the optimal allocation varies across datasets and tasks, and is not necessarily balanced. Our work builds on theirs in several ways, but differs crucially in that we propose a (theoretically-grounded) explanation for *why* subgroup allocation impacts subgroup performance so variably. Unlike Rolf et al. (2021), who must train many models across dataset sizes and allocations to fit their empirical scaling law, we identify an underlying mechanism driving these effects. This enables us to determine, for a given model and subgroup, whether subgroup allocation is likely to matter, and can help guide fine-tuning strategies to maximise subgroup performance.

## 3 PROBLEM SETTING

To study the impact of subgroup allocation on subgroup performance, we consider supervised fine-tuning of a pre-trained model on a dataset of input–label pairs $(x, y) \in \mathcal{X} \times \mathcal{Y}$. The data are drawn from an underlying distribution $\mathcal{P}$, which we randomly split into three disjoint subsets: $\mathcal{P}_{\text{pre-train}}$, $\mathcal{P}_{\text{fine-tune}}$, and $\mathcal{P}_{\text{test}}$. We study settings where the training and test distributions are annotated with $m$ binary attributes $\{A^{(1)}, \ldots, A^{(m)}\}$ which can represent demographic or other sample-level characteristics. Each attribute $A^{(j)}$ induces a binary partition of the data into two subgroups, $a_0^{(j)} = \{(x, y) \mid A^{(j)} = 0\}$ and $a_1^{(j)} = \{(x, y) \mid A^{(j)} = 1\}$. Examples of attributes include gender (male/female), imaging view (frontal/lateral), or dataset source (scanner A/scanner B).

For each subgroup, we record its base population prevalence under $\mathcal{P}$: $\gamma_k^{(j)} = \Pr_{(X,Y,A^{(j)}) \sim \mathcal{P}}[A^{(j)} = k]$, $k \in \{0, 1\}$. During fine-tuning, we investigate the impact of manipulating the prevalence of each subgroup, which we refer to as **subgroup allocation**. We assume there is a fixed fine-tuning budget of $K$ examples $\{(x_i, y_i, A_i^{(j)})\}_{i=1}^K$. For each subgroup $a_k^{(j)}$, following Rolf et al. (2021), we define its allocation as the fraction of the fine-tuning dataset coming from subgroup $a_k^{(j)}$:

$$\alpha_k^{(j)} := \frac{1}{K} \sum_{i=1}^K \mathbf{1}[A_i^{(j)} = k], \quad k \in \{0, 1\}.$$

Our objective is to characterise how subgroup-specific test performance (for instance the loss $\ell_k^{(j)}$) depends on these allocations. The central question of this work is thus: how does $\ell_k^{(j)}$ vary with $\alpha_k^{(j)}$ and why does this sensitivity differ across attributes and subgroups?

## 4   Current explanations are unreliable

We begin by systematically compare how subgroup allocation affects subgroup performance across various empirical settings. We explore whether existing hypotheses, for instance that under-represented subgroups will benefit from increased allocation, can explain the patterns we observe.

### 4.1   Experimental setup

**Model training with different allocations** We start with a baseline model trained on a random subset of the original dataset and subsequently fine-tune it on datasets for which we systematically vary the allocations $\alpha$ of different subgroups. For each attribute $A^{(j)}$, we partition the dataset into binary subgroups $g_0^{(j)}$ and $g_1^{(j)}$. For each attribute and dataset, we create 11 fine-tuning datasets, varying the allocation $\alpha_1^{(j)} \in \{0, 0.1, 0.2, \ldots, 1\}$ while keeping the total fine-tuning dataset size $K$ constant (ablations on $K$ are presented in Figure J25). Correspondingly, $\alpha_0^{(j)} = 1 - \alpha_1^{(j)}$. This yields, for instance, a dataset with 0% female images, 10% female images, and so on until 100%.

**Assessing sensitivity to subgroup allocation** We quantify how subgroup allocation affects model performance on each subgroup $g^j$ in two ways. First, we fit a linear least-squares regression to the subgroup-specific performance (e.g., accuracy, loss, AUC) as a function of the allocation $\alpha_k^{(j)}$, recording the slope $a_k^{(j)}$. We then average this across both groups to obtain a slope estimate $a^j$. We also obtain a coarse estimate of generalisation by subtracting model performance on subgroup $g_k^{(j)}$ at 0% allocation from its performance at 100% allocation: $\Delta \ell_k^{(j)} = \ell_k^{(j)}(\alpha_k^{(j)} = 1) - \ell_k^{(j)}(\alpha_k^{(j)} = 0)$.

**Datasets, tasks, and models** We conduct these experiments in four image and text datasets with a range of model architectures for binary classification tasks. This includes even/odd digit prediction with a red and green coloured version of **MNIST** (Lecun et al., 1998), pleural effusion classification in **MIMIC-CXR** (chest-X rays) (Johnson et al., 2019), skin lesion detection in **HAM10000** (skin images) (Tschandl et al., 2018), and toxic comment classification in **Civil_comments** (Borkan et al., 2019). These datasets all contain various metadata which enables natural splitting of the samples into subgroups, based on attributes like sex, ethnicity, image type, date of image etc. The multitude of attributes we compare within the same dataset allows us to gain more insights than previous studies which usually only consider one or two standard groupings (Rolf et al., 2021; Claucich et al., 2025; Idrissi et al., 2022). We use CNNs and transformers for our experiments. Detailed dataset characteristics and model implementation specifics are provided in Tables B1 and 2.

### 4.2   Sensitivity to subgroup allocation is highly variable

Across our three real-world datasets, we find substantial variability in subgroup performance sensitivity to allocation. Certain subgroups show no benefit from increased allocation and achieve equivalent performance whether the model is fine-tuned only on that subgroup or entirely without it (e.g., age in MIMIC). In contrast, other subgroups, like dataset of origin in HAM10000, are sensitive to their allocation. This results an estimated slopes of the accuracy change which range from 0 (no effect) to 0.05 (strong sensitivity), as shown in Figure 2. In other words, while the MIMIC model performs equivalently on old individuals whether or not it has been trained on such images, the HAM model is almost 10% less accurate on one dataset source if it has not been trained on any images from it. We summarise all slopes obtained in Tables C3 and C4. These results mirror other recent work which showed that while subgroup performance is sometimes maximised when a dataset is balanced with respect to a certain attribute, it can also be maximised at skewed allocations, or in other cases it can be equally maximised across allocations Rolf et al. (2021); Claucich et al. (2025); Roh et al. (2021); Čevora et al. (2025); Weng et al. (2023).

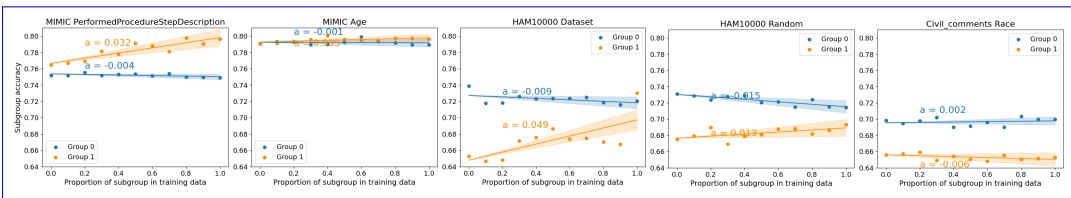

Figure 2: **While some subgroups' accuracy increases with increased representation in training data, others' performance is independent of their training data representation**. The fine-tuned model's balanced accuracy on each subgroup, averaged across 9 fine-tuning runs, is shown alongside estimated linear regression slopes $a$.

We consider more complex functional forms for fitting subgroup loss vs. allocation (e.g., power-law models (Rolf et al., 2021)), but we find them unstable: fits vary substantially with small changes in data and standard deviations are large, an issue also reported in Rolf et al. (2021)). We further discuss this in §C.2. We attribute this to small sample sizes and high heterogeneity across subgroups. In contrast, linear regression provides robust and interpretable summaries, and subgroup losses appear roughly linear across allocations. We therefore adopt linear fits as a first-order sensitivity measure.

### 4.3 CURRENT HYPOTHESES DO NOT EXPLAIN DIFFERENCES IN ALLOCATION SENSITIVITY

We explore common explanations for sensitivity to subgroup allocation including whether it could be linked to certain subgroups being under-represented in the initial pre-training dataset, certain subgroups being disadvantaged by the pre-trained model's performance, or certain subgroups having substantial class imbalances (Figures 3 and D8 respectively). However, none of these three explanations appear to be consistently correlated with sensitivity to subgroup allocation. For instance, we see certain subgroups which are extremely under-represented in the pre-training set (e.g., less than 20% of the pre-training data) which show no reduction in loss as fine-tuning data allocation increases (Figure 3 top row). This suggests that over-representing under-represented subgroups does not necessarily yield performance improvements, in line with other recent work (Schrouff et al., 2023; Claucich et al., 2025; Roh et al., 2021) and contradicting many other pieces of research (Idrissi et al., 2022; Wang & Russakovsky, 2023; Alabdulmohsin et al., 2024).

Similarly, we see surprising examples where certain subgroups which are initially amongst the lowest performing by the pre-trained model also show almost no decrease in loss as allocation increases (Figure 3 bottom row), again contradicting the general assumption that training on more data from a poor-performing subgroup will improve model performance on that subgroup (or improve it more than training on general data) (Roh et al., 2021; Sagawa et al., 2020).

## 5 BOUNDING SENSITIVITY TO SUBGROUP ALLOCATION WITH MODEL LATENT REPRESENTATIONS

Given the lack of a coherent explanation for differences in sensitivity to subgroup balance, we introduce the following hypothesis: sensitivity to subgroup allocation may be explained by whether a model learns distinct representations for subgroups $a_0$ and $a_1$, and thus needs to be trained on sufficiently high proportion of samples from, say $a_0$, in order to achieve good performance on $a_0$.

Through theoretical analysis and with certain assumptions, we are able to validate this hypothesis. We show that low latent separation with respect to an attribute $A$ implies low sensitivity to $A$ allocation in the fine-tuning dataset, i.e., fine-tuned model performance on $a_0$ and on $a_1$ does not significantly vary for different allocations $\alpha^{(A)}$. We formalise this idea by proving that small total variation distance (TV) between class-conditioned subgroup representations $Z$ in a pre-trained model, implies that last-layer fine-tuning on any dataset which differs in the allocation of $A$, but not in its proportion of $Y$, cannot result in models which differ significantly specifically in their subgroup accuracies (Theorem 5.1). To the best of our knowledge, this is the first result that links

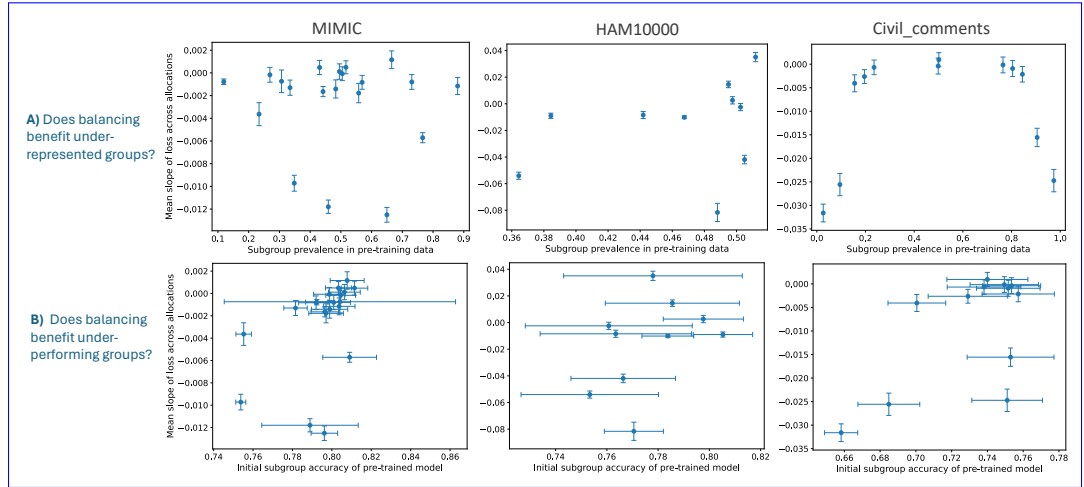

Figure 3: **Performance on subgroups under-represented** during pre-training (top) and performance on **disadvantaged subgroups** (bottom) does not necessarily improve with increasing dataset allocation. The y-axis shows the gradient of subgroup loss change with respect to subgroup allocation (negative values indicate performance improvement). No clear correlation is observed in either setting. Each point represents one subgroup, with error bars showing variation across 9 runs.

subgroup allocation sensitivity to class-conditional representation separation. For readability, we provide only proof sketches here, deferring the full derivations to Appendix E.

**Lemma 5.1.** *Let $f_{\theta,\eta}(x) = g_\theta(h_\eta(x))$ with representation $Z = h_\eta(X)$ and predictor $\hat{Y} = g_\theta(Z)$, where $g_\theta$ is the last layer. $\mathbb{P}_\eta$ denotes the distribution over representations $Z$. Assume that*

$$\mathsf{TV}\big(\mathbb{P}_\eta[Z \mid Y = y, A = 1], \mathbb{P}_\eta[Z \mid Y = y, A = 0]\big) \leq \varepsilon,$$

*for $y \in \{0, 1\}$. Then, it holds $\big|\mathbb{P}_\eta(Z \mid Y = y, A = a) - \mathbb{P}_\eta(Z \mid Y = y)\big| \leq \varepsilon$ for all $a \in \{0, 1\}$.*

This theorem tells us that if the representation $Z$ is similar between groups (i.e., $A = 0$ and $A = 1$) for a given label $Y = y$, then each group's representation is also close to the overall representation for that label, meaning group membership does not significantly affect the representation once the label is fixed. We can use this lemma to prove the main result. The proof is shown in E.1.

**Assumptions.** We assume that (i) fine-tuning datasets $\mathsf{D}'$, $\mathsf{D}''$ differ only in subgroup allocations $\alpha^{(A)}$, and (ii) the marginal label distribution $P(Y)$ and conditional distribution $P(Y \mid A)$ remain unchanged across datasets. These assumptions are further discussed in Appendix E.2.1.

**Theorem 5.1** (Group accuracy parity)**.** *Let $f_{\theta,\eta}(x) = g_\theta(h_\eta(x))$ with representation $Z = h_\eta(X)$ and predictor $\hat{Y} = g_\theta(Z)$, where $g_\theta$ is the last layer. For a dataset $\mathsf{D}$, define the quantity*

$$\mathsf{TV}(\mathsf{D}) := \mathbb{E}_y \left[ \mathsf{TV}\big(\mathbb{P}_\eta[Z \mid Y = y, A = 1], \mathbb{P}_\eta[Z \mid Y = y, A = 0]\big) \right].$$

*Suppose that the model is fine-tuned on two datasets $\mathsf{D}'$, $\mathsf{D}''$ which differ only in $\alpha^{(A)}$, yielding two models with parameters $\theta'$ and $\theta''$. If $\mathsf{TV}(\mathsf{D}') \leq \varepsilon$ and $\mathsf{TV}(\mathsf{D}'') \leq \varepsilon$, then*

$$|\mathrm{Acc}_{\theta'}(A = a) - \mathrm{Acc}_{\theta''}(A = a)| \leq 4\varepsilon + |\mathrm{Acc}_{\theta'} - \mathrm{Acc}_{\theta''}|$$

*for all $a \in \{0, 1\}$.*

This theorem tells us that if two models are fine-tuned on datasets that differ only in the group proportions (i.e., the distribution of $A$), and both models learn approximately group-invariant representations (i.e., $\mathsf{TV}(\mathsf{D}) \leq \varepsilon$), then the accuracy on any group $A = a$ will not differ much between the models.

The proof (E.1) uses a result that bounds how much the model's class-conditioned predictions can differ across groups using total variation distance (Lemma 5.1). Assuming that the label distribution $\mathbb{P}(Y)$ and the conditional distribution $\mathbb{P}(Y \mid A)$ remain unchanged across the datasets, this implies that the subgroup accuracy difference is bounded by a term depending on $\varepsilon$ and the difference in overall accuracies between the models.

**Remark (Scope of the last-layer assumption).** The statement and proof of Theorem 5.1 do not rely on retraining only the last layer. For any fixed representation map $h\colon \mathcal{X} \to \mathcal{Z}$ (including $h = \mathrm{Id}$ so that $Z = X$), if the class-conditional subgroup distributions of $Z$ have TV bounded by $\varepsilon$ then the same accuracy bound holds for any readout $g_\theta$ trained on $Z$. In practice, however, evaluating TV in the input or very early-layer spaces often yields large values and a vacuous bound.

**Remark (Tightness of bound).** In practice, for a fixed fine-tuning budget $K$, the overall accuracy difference is typically negligible relative to subgroup accuracy differences, i.e. $|\Delta\mathrm{ACC}| \ll |\Delta\mathrm{ACC}_a|$. We observe this in our experiments (Figure E9), implying that TV differences are the dominant driver in the upper bound.

# 6 SUBGROUP SEPARATION PREDICTS SENSITIVITY TO ALLOCATION IN REAL-WORLD EXPERIMENTS

We showed that if subgroup representations are nearly indistinguishable (as measured by TV), then modifying fine-tuning dataset subgroup allocation (assuming that $P(Y)$ and $P(Y|A)$ are unchanged) has little effect on downstream accuracies. We now turn to empirical analyses to test how well this theoretical upper bound captures real-world behaviour, and to investigate whether finer-grained patterns, such as correlations between representation separation and allocation sensitivity, emerge beyond what the bound alone reveals.

## 6.1 ASSESSMENT OF REPRESENTATION INVARIANCE

We keep the same setup as previously, where models are pre-trained on a random subset of each of the four datasets, and their last layer is then fine-tuned with varying subgroup allocations. To cover a broad range of subgroups, we relax the theorem's assumption that $P(Y)$ is fixed across fine-tuning distributions. Notably, many attributes do satisfy (or closely approximate) this assumption as they have equal class prevalences. This includes gender, marital status, language, and race in MIMIC; localisation in HAM10000; year and race in Civil_comments; and random groups across all datasets (full details listed in Table B2).

We quantify subgroup separation by extracting penultimate-layer embeddings, projecting them to a lower-dimensional space using PCA (retaining $\geq 70\%$ variance), and computing the mean **total variation** distance (TV) between subgroup distributions conditioned on $Y$. TV is bounded in $[0, 1]$, with higher values indicating stronger separation. For completeness, we also explore additional distance metrics including the **Wasserstein distance** (WD) and the **Fréchet distance** (FD) which emphasise distinct representation differences. We give additional methodological details, show that our metrics are robust to whether they are calculated on the full feature-space or the reduced feature space, and show that all three distances metrics are correlated in Appendix Section F.

## 6.2 INTUITION FOR OUR RESULTS IN MNIST

In our synthetic MNIST set-up, we *know* that a good parity classification model should not rely on colour ($A^{(0)}$) to make a prediction, so learnt representations $z$ should be invariant to $A^{(0)}$, in other words $P(z \mid Y, a_0^{(0)}) = P(z \mid Y, a_1^{(0)})$. In contrast, a model must encode some notion of which digit is represented in order to classify it into even vs. odd, and therefore its learned representation *should* depend on whether the digit is over 5 or under 5 ($A^{(1)}$). We test this by training a 2-layer CNN. As expected, the penultimate layer model embeddings do cluster by digit $A^{(1)}$ but not by colour $A^{(0)}$ (Figure 1). Quantitatively, the average TV between the images representing digits over 5 and those under 5 is 0.17, approximately twice the average distance between the red and green images (which is itself close to that between random groupings), as shown in Figure F11.

Our hypothesis predicts that fine-tuning this model on datasets with different proportions of the same subgroups will show that the model is only sensitive to the allocation of the under 5/over 5 groups but not to the red/green groups. Indeed, we find that the subgroup accuracy on under 5 and over 5 images drops sharply as their fine-tuning dataset allocation drops, while the subgroup accuracy in the red/green images is roughly independent of the fine-tuning dataset allocation (Figure 1, bottom row). The model can effectively generalise "zero-shot" to new colours, but not to unseen digit

groups. Looking back at common explanations in the literature, we note that both red/green and over 5/under 5 groups have equal $P(Y)$, equal base accuracy, and equal base allocation in the pre-training dataset, so none of those explanations would have been able to predict the observed discrepancy in subgroup allocation sensitivity.

### 6.3 SUBGROUP SEPARATION IS HIGHLY CORRELATED TO ALLOCATION SENSITIVITY

We next explore how our hypothesis holds in real-world datasets with more subgroups and larger models, where there is no clear-cut separation between attributes a model should be invariant to or not. We first note that there is wide variation in subgroup separation in penultimate-layer representations (as shown in Figure 11). Generally, image-related attributes (e.g., X-ray view in MIMIC or dataset of origin in HAM10000) induce greater separation than demographic attributes. Some subgroups, such as Civil_comments year (pre- vs. post-2017) or HAM10000 lesion location (extremity vs. trunk), show separations comparable to random splits. The three distance metrics (TV, WD, FD) are strongly correlated and robust across full vs. PCA-reduced spaces (Figures F12 and 13).

Across our three real-world datasets (image and text) and multiple architectures (CNNs and transformers), we find a significant correlation between subgroup separation in pre-trained representations and subgroup sensitivity to allocation during fine-tuning (Figure 4). This correlation holds across the three distance metrics (TV, WD, FD). Intuitively, when two subgroups have similar representation of target Y (i.e., $P(z \mid Y, a_0^{(0)}) = P(z \mid Y, a_1^{(0)})$), then additional subgroup specific data provides little added value. Conversely, when subgroup representations are separated, allocation has a large effect.

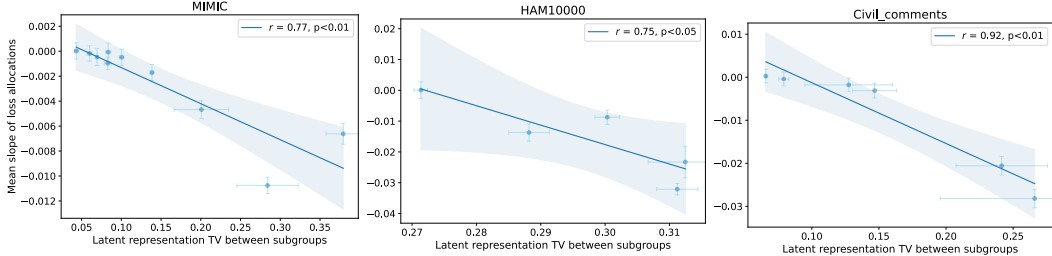

Figure 4: **Sensitivity to subgroup allocation is highly correlated with separation** in the pre-trained model's representation space (as measured by total variation distance, TV) across the three datasets. Each dot represents mean TV and loss slope for one subgroup, averaged across 9 fine-tuning runs, with bars corresponding to standard deviations, and Pearson correlation also shown.

We also observe that zero-shot generalisation is directly related to subgroup separation. As shown in Figure 5, subgroup AUC is constant between 100% and 0% training data allocation only when representations are (approximately) invariant across them. This echoes work in the domain adaptation literature, which stipulate that invariant representations are more robust as they generalise across environments (Arjovsky et al., 2020). Similar ideas have also laid the foundation for "fair representation learning" methods which aim to mitigate biases by encouraging models to learn causal representations of $Y$ rather than encoding, e.g., demographic attributes like sex (Sarhan et al., 2020; Madras et al., 2018). Here, we extend this principle to subgroup balancing by using a measurable property of the *pre-trained* model, class-conditional latent separation, as a predictive diagnostic for when allocation will matter. We further provide a bound (Theorem 5.1) linking small total-variation separation to small subgroup-accuracy differences across allocations. Altogether, these empirical results support our theoretical finding of an upper bound to performance differences across allocations, and the consistent correlation suggests there may even be a stronger phenomenon at play.

While we first experiment with last-layer fine-tuning for consistency with our theory, we test whether this correlation extends to settings where the fine-tuning occurs on the full network (and therefore representations $Z$ could change). We find equally significant correlations, with effects of stronger magnitudes, across all three datasets (Figure H17 and H18), suggesting that this trend may hold in less restricted settings. Surprisingly, analysis of the separation of last-layer representations shows that they are roughly constant across allocations, effectively matching our initial setup (Figure H16).

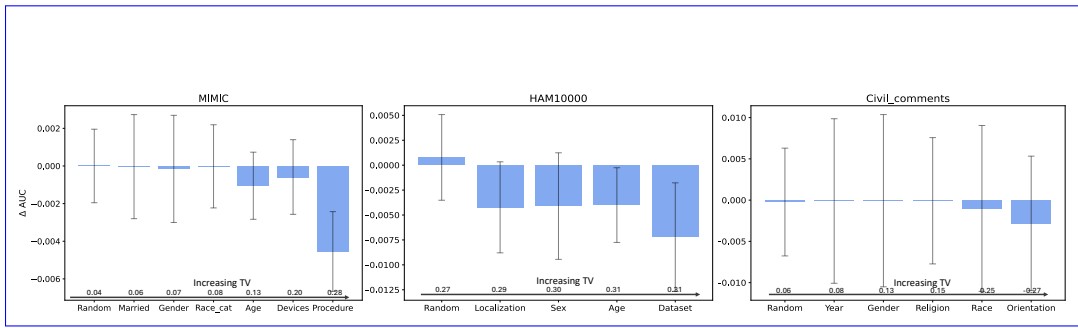

Figure 5: **AUC difference when a model has not been trained on a subgroup increases with the separation of the latent representations of the subgroups.** Results are shown as mean AUC difference across 9 fine-tuning runs with error bars indicating standard deviation.

We also test whether our results hold when training from scratch (using the model trained on the natural dataset proportions to analyse latent representation separation), and find that the same trends persist, with allocation sensitivities almost 10 times stronger (full results in Appendix H.2).

## 6.4 OPERATIONALISATION OF LATENT SEPARATION HYPOTHESIS

We further test our hypothesis by explicitly *enforcing* low TV (via a differentiable proxy) during pre-training to see whether this affects sensitivity to subgroup allocation (method detailed in § J). We find that regularisation indeed reduces latent TV separation, which leads to a significant drop in overall performance (over 0.10 accuracy decrease) but reduced performance gaps between frontal and lateral images, and importantly, reduced sensitivity to subgroup allocation (accuracy slope decreases from 0.016 to -0.007 with regularisation), as shown in Figure 6. This intervention directly supports our hypothesis that latent separation *drives* allocation sensitivity.

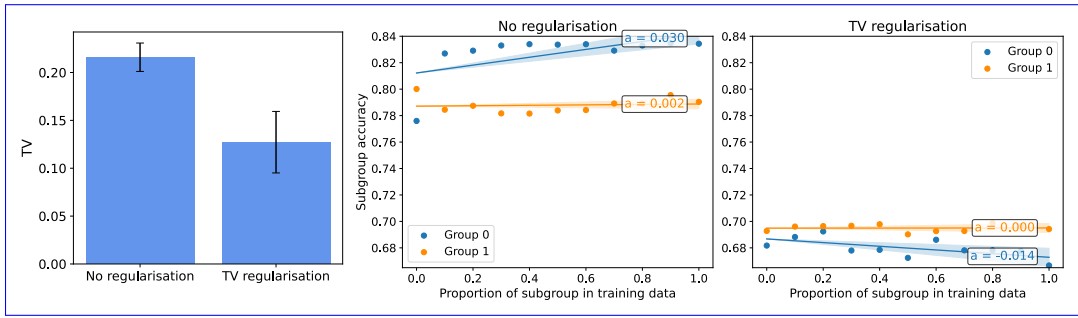

Figure 6: **TV regularisation reduces subgroup allocation sensitivity.** Mean latent TV distance between view-groups after pre-training (left), fine-tuned subgroup balanced accuracy as subgroup allocation increases without TV regularisation (middle) and with regularisation (right).

## 6.5 PRACTICAL APPLICATION IN FINE-TUNING A FOUNDATION MODEL

Beyond our findings' analytical value in explaining previously observed discrepancies in subgroup allocation sensitivity, we also examine its practical utility. Specifically, we consider a more realistic setup where a foundation model (FM) is used to generate informative embeddings of a dataset, on top of which a simple classification layer is trained. For this, we use two radiology-specific FMs: CheXagent (Chen et al., 2024) and RAD-DINO-MAIRA-2 (Pérez-García et al., 2024). Both are trained on over 1 million images, with distinct training mechanisms (e.g., image and text via SigLIP and only images via DINO respectively). We use these two models to embed images in the MIMIC-CXR test set. This differs from our traditional setup as the pre-training and fine-tuning datasets are drawn from different distributions and the fine-tuning task does not directly overlap with the pre-training tasks.

We again measure distances between subgroup representations and find substantial variability (Figures I21). Interestingly, demographic attributes such as gender and age are more separated in the FM than in our task-specific model, likely because a general-purpose model encodes broad features rather than only task-specific ones. Across both models, the most separated groups remain imaging-related, followed by age, gender, and ethnicity. Based on our hypothesis, we predict that sensitivity to subgroup allocation should follow this ordering, and that if we are concerned about maximising subgroup performance across each of these attributes, we should prioritise balancing the dataset with respect to imaging variables. Indeed, our experiments confirm this: balancing by gender or race has little effect, but balancing by X-ray procedure or view significantly increases mean subgroup accuracy when using a 50/50 allocation compared to 100/0 in both models (Figure 7, I22). Full analysis of allocation sensitivity and latent separation shows

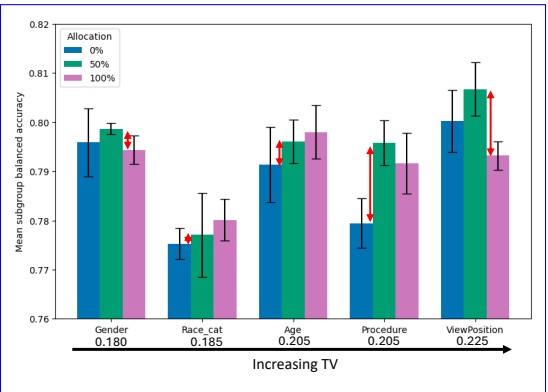

Figure 7: **In foundation model fine-tuning**, selecting a balanced allocation for imaging subgroups increases subgroup accuracy by over 0.02, but has less importance for demographic groups, as predicted by their reduced total variation distance. We show results of 3 fine-tuning runs, with red arrows for potential balancing gains.

strong correlations ($r \in [0.60, 0.95]$, Figure I23, 24), supporting our hypothesis under much weaker assumptions and highlighting its potential practical implications for data curation and model fine-tuning. Concretely, practitioners could use this TV calculation to prioritise obtaining balanced data with respect to procedure and image view in further fine-tuning data collection efforts.

### 6.6 LIMITATIONS AND FUTURE WORK

The core of our study is restricted to a pre-training–then–fine-tuning setup, as this is necessary to first explore representation separation before deciding how to allocate data. One consequence is that the effects we observe are often modest in magnitude, but they remain consistent and statistically significant across the different random seeds, data splits, and pre-trained checkpoints we use. We also do not provide a method for determining the optimal allocation strategy to maximise accuracy across multiple subgroups. We caution that precise recommendations (e.g., targeting 40/60 female/male or 20/80 young/old ratios) would likely be unreliable due to factors like confounders, noise, or differences in sample informativeness, and in any case difficult to operationalise. Instead, we argue that a broader understanding of *whether* and *how strongly* subgroup allocation influences performance is more relevant to current concerns in fairness and domain generalisation. Moreover, future work should explore how our findings apply to more complex settings, including tasks other than binary classification, with multi-valued or even continuous subgroups, and settings where the training data is expanded instead of simply re-allocated. Future work should also explore if our findings can be leveraged for bias mitigation (for instance by enforcing representation invariance where appropriate). Finally, our work is limited, as in all fairness research, by our use of subgroups. For example, finding that a model is insensitive to the allocation of white vs. non-white samples does not imply that allocation across ethnic groups as a broad concept is irrelevant, simply that, when described by our chosen coarse groupings, it does not matter.

### 7 CONCLUSIONS

We propose, prove (under certain assumptions), and give extensive empirical evidence for, a novel hypothesis explaining why in some cases subgroup training data allocation does not matter for subgroup performance, and in other cases, it is crucial. Unlike standard explanations (e.g., assuming that under-represented or poorly performing-groups always benefit from increased data representation), our hypothesis consistently matches empirical results across diverse datasets and models. By predicting subgroup allocation sensitivity through latent representation analysis, we provide a new way to inform crucial training data decisions, maximising fairness, accuracy, and efficiency.

## 8 REPRODUCIBILITY STATEMENT

We provide extensive details in the appendix to ensure reproducibility of our results. For each dataset, we describe pre-processing steps, model architectures, hyper-parameter optimisation procedures, and training details. We pre-train three models with different random seeds per dataset and conduct fine-tuning across three more random data splits, yielding nine runs per allocation. We further conduct ablations on different representation distance metrics, full vs. last-layer fine-tuning, and fine-tuning budget $K$. We release code and scripts (upon acceptance) to replicate all figures and analyses.

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

# A  APPENDIX STRUCTURE

- § B: Supplementary details on datasets, models, and their implementations
- § C: Supplementary results on sensitivity to subgroup allocation and ways to model it
- § D: Supplementary results on common hypotheses not holding
- § E: Full theoretical results and further remarks
- § F: Supplementary results on latent representation distances
- § G: Supplementary results on the correlation between representation distance and subgroup allocation sensitivity
- § H: Extending setup to other training regimes (full fine-tuning and training from scratch)
- § I: Supplementary results on foundation model fine-tuning
- § J: Operationalisation of hypothesis through TV regularisation
- § J: Ablations on fine-tuning budget $K$ in MIMIC
- § L: LLM Usage

# B  SUPPLEMENTARY EXPERIMENTAL DETAILS

We present additional details on the four datasets and models used. We do not use the full datasets for fine-tuning because to keep all fine-tuning allocation datasets the same size we set them to the size of the smallest subgroup in each dataset. For each dataset, we pre-train three models with different random seeds. For each subgroup allocation experiment, we fine-tune the final classification layer of each pre-trained model on the three randomly generated data splits, resulting in a total of nine fine-tuned models per subgroup allocation. This is then repeated across all the subgroups we consider. This reflects one of the strengths of our work, that compared to the typical data balancing paper, we compare many different subgroups within a dataset (5 to 11), allowing us to gain new insights on why they may have different properties.

Table 1 summarises the implementation choices for each dataset. We conduct hyperparameter tuning both for pre-training and fine-tuning. We optimise hyperparameters within the following ranges:

- **Batch size**: 64, 128, 256
- **Learning rate**: [1e-6:1e-3]
- **Weight decay**: [1e-6:1e-2]

Table 1: Implementation details for all models.

| Training strategy | MNIST | MIMIC-CXR | HAM10000 | Civil_comments |
|---|---|---|---|---|
| $Y$ | Even/odd digit | Pleural effusion | Malignant/benign lesion | Comment toxicity |
| **Backbone** | 2-layer CNN | DenseNet121 Huang et al. (2016) | ViTB16 Dosovitskiy et al. (2021) | BERTClassifier (uncased) Devlin et al. (2018) |
| **Initialisation** | None | ImageNet Deng et al. (2009) | ImageNet Deng et al. (2009) | Bookcorpus, Wikipedia (English) |
| $N_{pretrain}$ | 5,000 | 33,237 | 2,000 | 17,920 |
| $N_{finetune}$ | 24,000 | 22,353 | 3,184 | 24,000 |
| $N_{test}$ | 4,000 | 26,590 | 1,000 | 12,243 |
| **Image size** | 3x28x28 | 3x256x256 | 3x256x256 | NA |
| **Augmentation** | Flip, rotation, Gaussian blur | Flip, rotation, affine transformation, crop | Flip, rotation, color jitter, affine transformation, crop | None |
| **Optimiser** | Adam | Adam | AdamW | AdamW |
| **Loss** | Binary cross-entropy | Binary cross-entropy | Binary cross-entropy | Binary cross-entropy |

We create natural subgroups based on the metadata available, such as based on sex, age, ethnicity, or image-related attributes like the position of the X-ray. subgroups are generated based on whether the comment mentions specific attributes, like gender, religion, or sexual orientation.

Table 2: Subgroups used for each dataset and their base population prevalence and class prevalence. Subgroups which satisfy our main theorem's assumptions are highlighted in green.

| Dataset | Attribute | Group 0 / 1 | P(A=1) | P(Y\|A=0)/P(Y\|A=1) |
|---|---|---|---|---|
| **MNIST** | Colour | Red / Green | 0.50 | 0.50 / 0.50 |
| | Digit value | Below 5 / Over | 0.50 | 0.50 / 0.50 |
| | Random | / | 0.50 | 0.50 / 0.50 |
| **MIMIC-CXR** | View Position | Lateral / Frontal | 0.65 | 0.16 / 0.26 |
| | Patient Orientation | Recumbent / Erect | 0.77 | 0.35 / 0.20 |
| | Procedure | Portable / Fixed scanner | 0.46 | 0.35 / 0.11 |
| | Support_Devices | Absent / Present | 0.23 | 0.17 / 0.39 |
| | Gender | Female / Male | 0.51 | 0.20 / 0.25 |
| | Insurance | Not-private / Private | 0.27 | 0.24 / 0.19 |
| | Language | English / Non-English | 0.88 | 0.22 / 0.22 |
| | Marital Status | Married / Unmarried | 0.44 | 0.20 / 0.25 |
| | Race | White / Non-White | 0.66 | 0.17 / 0.25 |
| | Age | Under 60 / Above 60 | 0.56 | 0.16 / 0.28 |
| | Random | / | 0.50 | 0.22 / 0.23 |
| **HAM10000** | Sex | Male / Female | 0.54 | 0.11 / 0.22 |
| | Age | Under 50 / Above 50 | 0.46 | 0.08 / 0.27 |
| | Dataset of origin | Rosendahl or Vidir_molemax / Neither | 0.62 | 0.23 / 0.13 |
| | Localisation of lesion | Central / Extremity | 0.38 | 0.16 / 0.17 |
| | Random | / | 0.53 | 0.16 / 0.17 |
| **Civil_comments** | Gender | No mention / Mention | 0.20 | 0.10 / 0.15 |
| | Orientation | No mention / Mention | 0.03 | 0.11 / 0.26 |
| | Religion | No mention / Mention | 0.16 | 0.11 / 0.13 |
| | Race | No mention / Mention | 0.09 | 0.10 / 0.26 |
| | Year of comment | Pre-2017 / Post-2017 | 0.76 | 0.11 / 0.11 |
| | Random | / | 0.50 | 0.11 / 0.11 |

## C    SUPPLEMENTARY RESULTS ON SUBGROUP ALLOCATION SENSITIVITY

### C.1    SENSITIVITY OF ALL ATTRIBUTES

We report full results for subgroup allocation sensitivity across all attributes for two full fine-tuning and last-layer fine-tuning (Tables 3 and 4 respectively).

Table 3: Sensitivity of subgroup performance to fine-tuning allocation across datasets and attributes in full model fine-tuning. Reported values are esimated regression slopes (mean $\pm$ standard deviation across nine fine-tuning runs) for subgroup loss, balanced accuracy, and AUC as allocation varies from 0% to 100%. Larger absolute slopes indicate higher sensitivity.

| Dataset | Attribute | Loss slope | Balanced Acc slope | AUC slope |
|---|---|---|---|---|
| MIMIC | View Position | -0.027 (0.001) | 0.034 (0.003) | 0.012 (0.001) |
| MIMIC | Patient Orientation | -0.009 (0.001) | 0.013 (0.002) | 0.003 (0.000) |
| MIMIC | Procedure | -0.026 (0.001) | 0.026 (0.002) | 0.016 (0.001) |
| MIMIC | Support_Devices | -0.009 (0.001) | 0.010 (0.001) | 0.006 (0.000) |
| MIMIC | Gender | -0.003 (0.001) | 0.004 (0.002) | 0.002 (0.000) |
| MIMIC | Insurance | -0.002 (0.001) | 0.003 (0.002) | 0.001 (0.000) |
| MIMIC | Language | -0.002 (0.001) | 0.001 (0.002) | 0.001 (0.000) |
| MIMIC | Marital_Status | -0.001 (0.001) | -0.002 (0.001) | 0.000 (0.000) |
| MIMIC | Race_cat | -0.002 (0.001) | -0.003 (0.002) | 0.001 (0.000) |
| MIMIC | Age | -0.006 (0.001) | 0.009 (0.002) | 0.003 (0.000) |
| MIMIC | Random | -0.000 (0.001) | 0.000 (0.002) | 0.000 (0.000) |
| HAM10000 | Sex | -0.090 (0.008) | 0.058 (0.003) | 0.032 (0.001) |
| HAM10000 | Age | -0.157 (0.011) | 0.049 (0.003) | 0.044 (0.001) |
| HAM10000 | Dataset | -0.230 (0.011) | 0.129 (0.003) | 0.077 (0.002) |
| HAM10000 | Localization | -0.173 (0.010) | 0.082 (0.003) | 0.049 (0.001) |
| HAM10000 | Random | 0.018 (0.008) | -0.010 (0.003) | -0.002 (0.001) |
| Civil_comments | Gender | -0.034 (0.011) | 0.023 (0.003) | 0.017 (0.003) |
| Civil_comments | Orientation | -0.243 (0.012) | 0.094 (0.004) | 0.089 (0.003) |
| Civil_comments | Religion | -0.048 (0.012) | 0.031 (0.003) | 0.025 (0.003) |
| Civil_comments | Race | -0.146 (0.014) | 0.037 (0.002) | 0.041 (0.002) |
| Civil_comments | Year | -0.022 (0.013) | 0.013 (0.003) | 0.011 (0.004) |
| Civil_comments | Random | -0.014 (0.011) | 0.008 (0.003) | 0.007 (0.003) |

### C.2    EXPERIMENTING WITH MORE COMPLEX FITS

We also experiment with the subgroup allocation scaling law model introduced by Rolf et al. (2021), which describes subgroup loss as a function of group size and total sample size:

$$\ell_g(n_g, n) \; = \; \sigma_g^2 \, n_g^{-p_g} \; + \; \tau_g^2 \, n^{-q_g} \; + \; \delta_g,$$

where $n_g$ is the number of samples in subgroup $g$, $n$ is the total number of samples, and $\sigma_g, \tau_g > 0$, $p_g, q_g > 0$. Following their setup, we conduct additional experiments where we vary the fine-tuning dataset size $K$ (to obtain more data points) and fit this model under the same parameter constraints.

In our experiments, however, we find that the fitted parameters often come with very large standard deviations, suggesting instability in the estimates. We also find that estimated parameters that are highly non-robust to small changes in how the fit is estimated or which data points are included. An example for two groups in MIMIC is given in Table 5. While we recognise that linear fits are imperfect, and do not always precisely capture subgroup performance at extreme allocations (e.g., 0 or 100), we use them in our main analysis as they reliably and consistently capture the first-order pattern, telling us whether performance changes across allocations and how much it changes. This is sufficient to capture a strong and consistent relationship to latent representation separation.

918
919
920
921
922
923
924
925
926
927
928

Table 4: Sensitivity of subgroup performance to fine-tuning allocation across datasets and attributes in last-layer fine-tuning. Reported values are esimated regression slopes (mean $\pm$ standard deviation across nine fine-tuning runs) for subgroup loss, balanced accuracy, and AUC as allocation varies from 0% to 100%. Larger absolute slopes indicate higher sensitivity.

| Dataset | Attribute | Loss slope | Balanced Acc slope | AUC slope |
|---|---|---|---|---|
| MIMIC | View Position | -0.007 (0.001) | 0.001 (0.002) | -0.000 (0.000) |
| MIMIC | Patient Orientation | -0.008 (0.001) | -0.002 (0.001) | 0.000 (0.000) |
| MIMIC | Procedure | -0.011 (0.001) | 0.014 (0.002) | 0.003 (0.000) |
| MIMIC | Support_Devices | -0.005 (0.001) | 0.000 (0.001) | 0.000 (0.001) |
| MIMIC | Gender | -0.000 (0.001) | 0.001 (0.002) | 0.000 (0.000) |
| MIMIC | Insurance | -0.000 (0.001) | 0.000 (0.002) | -0.000 (0.000) |
| MIMIC | Language | -0.001 (0.001) | -0.000 (0.001) | 0.001 (0.000) |
| MIMIC | Marital_Status | -0.000 (0.001) | -0.000 (0.002) | -0.000 (0.000) |
| MIMIC | Race_cat | -0.000 (0.001) | -0.002 (0.002) | -0.000 (0.000) |
| MIMIC | Age | -0.002 (0.001) | 0.002 (0.002) | 0.001 (0.000) |
| MIMIC | Random | 0.000 (0.001) | 0.001 (0.002) | -0.000 (0.000) |
| HAM10000 | Sex | -0.009 (0.002) | 0.004 (0.003) | 0.005 (0.001) |
| HAM10000 | Age | -0.032 (0.002) | -0.011 (0.003) | 0.004 (0.001) |
| HAM10000 | Dataset | -0.023 (0.005) | 0.020 (0.003) | 0.010 (0.001) |
| HAM10000 | Localization | -0.014 (0.003) | 0.010 (0.003) | 0.004 (0.001) |
| HAM10000 | Random | 0.000 (0.003) | -0.001 (0.003) | -0.000 (0.001) |
| Civil_comments | Gender | -0.002 (0.002) | 0.001 (0.002) | 0.000 (0.002) |
| Civil_comments | Orientation | -0.026 (0.002) | -0.000 (0.002) | 0.002 (0.002) |
| Civil_comments | Religion | -0.003 (0.002) | -0.000 (0.002) | 0.000 (0.002) |
| Civil_comments | Race | -0.022 (0.002) | 0.004 (0.002) | 0.000 (0.002) |
| Civil_comments | Year | -0.000 (0.002) | 0.000 (0.003) | 0.000 (0.002) |
| Civil_comments | Random | -0.000 (0.002) | -0.001 (0.002) | -0.000 (0.002) |

949
950
951
952
953
954
955
956
957
958
959
960
961

Table 5: Estimated power-law scaling model parameter fits and standard deviations.

| Attribute | Group | $\sigma_g$ | $p_g$ | $\tau_g$ | $q_g$ | $\delta_g$ |
|---|---|---|---|---|---|---|
| Gender | 0 | $0.000 \pm 0.000$ | $1.298 \pm 0.000$ | $3136.744 \pm 13745.833$ | $2.000 \pm 0.964$ | $0.302 \pm 0.020$ |
| Gender | 1 | $278.269 \pm 647.492$ | $2.000 \pm 0.650$ | $3002.106 \pm 13853.293$ | $2.000 \pm 1.015$ | $0.348 \pm 0.019$ |
| ViewPosition | 0 | $4.843 \pm 6.518$ | $0.850 \pm 0.387$ | $1.523 \pm 4.728$ | $0.225 \pm 1.188$ | $0.000 \pm 1.388$ |
| ViewPosition | 1 | $1.035 \pm 0.806$ | $0.260 \pm 0.336$ | $2.175 \pm 8.930$ | $0.276 \pm 1.359$ | $0.000 \pm 1.609$ |

967
968
969
970
971

# D SUPPLEMENTARY RESULTS ON COMMON HYPOTHESES NOT HOLDING

We additionally test the hypothesis that subgroups with a high class imbalance may lead to greater model sensitivity to their allocation by modifying $\mathcal{P}(Y)$. However, we see no consistent correlation between the two variables.

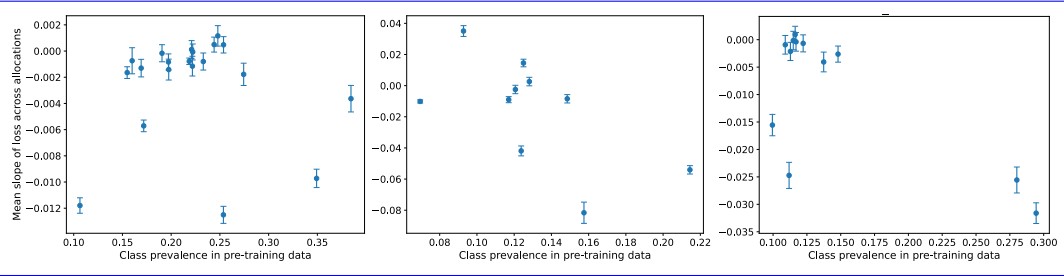

Figure 8: Performance on subgroups with high class imbalance during pre-training does not necessarily improve with increasing training data allocation. Each dot represents one subgroup with bars indicating variation across 9 fine-tuning runs.

# E  FULL THEORETICAL RESULTS

## E.1  FULL PROOFS

**Assumptions E.1.** *Throughout this section, we impose the following conditions:*

1. *Fine-tuning datasets D differ in subgroup allocations $\alpha^{(A)}$, but not in their overall label distribution $P(Y)$ or conditional distribution $P(Y \mid A)$.*

2. *Fine-tuning is restricted to the last-layer classifier $g_\theta$, with representation $Z$ held fixed.*

3. *We assume that all classifiers are Bayes optimal under realisability.*

**Lemma E.1.** *Let $f_{\theta,\eta}(x) = g_\theta(h_\eta(x))$ with representation $Z = h_\eta(X)$ and predictor $\hat{Y} = g_\theta(Z)$, where $g_\theta$ is the last layer. $\mathbb{P}_\eta$ denotes the distribution over representations $Z$. Assume that*

$$\mathsf{TV}\big(\mathbb{P}_\eta[Z \mid Y = y, A = 1], \, \mathbb{P}_\eta[Z \mid Y = y, A = 0]\big) \leq \varepsilon,$$

*for $y \in \{0, 1\}$. Then, it holds $\big|\mathbb{P}_\eta(Z \mid Y = y, A = a) - \mathbb{P}_\eta(Z \mid Y = y)\big| \leq \varepsilon$ for all $a \in \{0, 1\}$.*

*Proof.* We only prove the case for $A = 1$, since the case for $A = 0$ is analogous. For simplicity, write

$$\mu_a(\cdot \mid y) := \mathbb{P}_\eta\big(Z \mid Y = y, A = a\big)$$
$$\mu(\cdot \mid y :) = \mathbb{P}_\eta\big(Z \mid Y = y\big)$$
$$\pi_y := \mathbb{P}_\eta(A = 1 \mid Y = y)$$

By the law of total probability, we have that

$$\mu(\cdot \mid y) = \pi_y \, \mu_1(\cdot \mid y) + (1 - \pi_y) \, \mu_0(\cdot \mid y). \tag{1}$$

View $\mu_0(\cdot \mid y)$ and $\mu_1(\cdot \mid y)$ as probability measures on the same measurable space. By using equation 1, we have that

$$\mu_1(\cdot \mid y) - \mu(\cdot \mid y) = \mu_1(\cdot \mid y) - \big(\pi_y \mu_1(\cdot \mid y) + (1 - \pi_y)\mu_0(\cdot \mid y)\big) = (1 - \pi_y)\big(\mu_1(\cdot \mid y) - \mu_0(\cdot \mid y)\big).$$

Taking total variation norms and using homogeneity of total variation for signed measures,

$$\mathrm{TV}\big(\mu_1(\cdot \mid y), \mu(\cdot \mid y)\big) = (1 - \pi_y)\, \mathrm{TV}\big(\mu_1(\cdot \mid y), \mu_0(\cdot \mid y)\big).$$

By the defining property of total variation,

$$\sup_z \big|\mu_1(z \mid y) - \mu(z \mid y)\big| = \mathrm{TV}\big(\mu_1(\cdot \mid y), \mu(\cdot \mid y)\big) \leq (1 - \pi_y)\varepsilon \leq \varepsilon,$$

where we have used that $1 - \pi_y \leq 1$. Hence, it holds $\big|\mathbb{P}_\eta(Z \mid Y = y, A = 1) - \mathbb{P}_\eta(Z \mid Y = y)\big| \leq \varepsilon$, as claimed. $\qquad\square$

We can use this lemma to prove the main result.

**Theorem E.1** (Group accuracy parity). *Let $f_{\theta,\eta}(x) = g_\theta(h_\eta(x))$ with representation $Z = h_\eta(X)$ and predictor $\hat{Y} = g_\theta(Z)$, where $g_\theta$ is the last layer. For a dataset D, define the quantity*

$$\mathsf{TV}(\mathtt{D}) := \mathbb{E}_y \left[ \mathsf{TV}\big(\mathbb{P}_\eta[Z \mid Y = y, A = 1], \, \mathbb{P}_\eta[Z \mid Y = y, A = 0]\big) \right].$$

*Suppose that the model is fine-tuned on two balanced datasets $\mathtt{D}'$, $\mathtt{D}''$ with different proportions of the attribute $A$, yielding parameters $\theta'$ and $\theta''$. If $\mathsf{TV}(\theta') \leq \varepsilon$ and $\mathsf{TV}(\theta'') \leq \varepsilon$ [1], then*

$$|\mathrm{ACC}_{\theta'}(A = a) - \mathrm{ACC}_{\theta''}(A = a)| \leq 4\varepsilon + |\mathrm{ACC}_{\theta'} - \mathrm{ACC}_{\theta''}|$$

*for all $a \in \{0, 1\}$.*

---

[1] If the TVs are bounded by different constants, e.g., $\varepsilon$ and $\delta$, then the upper bound can be rewritten as: $2\varepsilon + 2\delta + |\mathrm{ACC}_{\theta'} - \mathrm{ACC}_{\theta''}|$.

*Proof.* We can write the accuracy via conditioning on the true label as

$$\text{ACC}_{\theta'}(A = a) = \mathbb{P}_{\theta'}(\hat{Y} = Y \mid A = a) = \sum_y \mathbb{P}_{\theta'}(\hat{Y} = y \mid Y = y, A = a)\,\mathbb{P}_{\theta'}(Y = y \mid A = a) \tag{2}$$

and

$$\text{ACC}_{\theta''}(A = a) = \mathbb{P}_{\theta''}(\hat{Y} = Y \mid A = a) = \sum_y \mathbb{P}_{\theta''}(\hat{Y} = y \mid Y = y, A = a)\,\mathbb{P}_{\theta''}(Y = y \mid A = a) \tag{3}$$

We know by construction that class-conditional label distributions match, i.e.,

$$\mathbb{P}_{\theta'}(Y = y \mid A = a) = \mathbb{P}_{\theta''}(Y = y \mid A = a) =: p_{a,y} \tag{4}$$

Combining equation 2-equation 4 it holds

$$|\text{ACC}_{\theta'}(A = a) - \text{ACC}_{\theta''}(A = a)| = \left| \sum_y \mathbb{P}_{\theta'}(\hat{Y} = y \mid Y = y, A = a)\,p_{a,y} - \sum_y \mathbb{P}_{\theta''}(\hat{Y} = y \mid Y = y, A = a)\,p_{a,y} \right|$$

$$\leq \sum_y \left| \mathbb{P}_{\theta'}(\hat{Y} = y \mid Y = y, A = a) - \mathbb{P}_{\theta''}(\hat{Y} = y \mid Y = y, A = a) \right| |p_{a,y}|$$

$$\leq \sum_y \left| \mathbb{P}_{\theta'}(\hat{Y} = y \mid Y = y, A = a) - \mathbb{P}_{\theta''}(\hat{Y} = y \mid Y = y, A = a) \right|, \tag{5}$$

where we have used the triangle inequality, together with the fact that $0 \leq p_{a,y} \leq 1$. Furthermore, using the triangle inequality and Lemma 5.1[2], it holds

$$\sum_y \left| \mathbb{P}_{\theta'}(\hat{Y} = y \mid Y = y, A = a) - \mathbb{P}_{\theta''}(\hat{Y} = y \mid Y = y, A = a) \right|$$

$$\leq \sum_y \left| \mathbb{P}_{\theta'}(\hat{Y} = y \mid Y = y, A = a) - \mathbb{P}_{\theta'}(\hat{Y} = y \mid Y = y) \right|$$

$$+ \left| \sum_y \mathbb{P}_{\theta'}(\hat{Y} = y \mid Y = y) - \mathbb{P}_{\theta''}(\hat{Y} = y \mid Y = y) \right|$$

$$+ \sum_y \left| \mathbb{P}_{\theta''}(\hat{Y} = y \mid Y = y, A = a) - \mathbb{P}_{\theta''}(\hat{Y} = y \mid Y = y) \right|$$

$$\leq 4\varepsilon + \left| \sum_y \mathbb{P}_{\theta'}(\hat{Y} = y \mid Y = y) - \mathbb{P}_{\theta''}(\hat{Y} = y \mid Y = y) \right| \tag{6}$$

The claim follows by combining equation 5 with equation 6, and noting that since the parameters $\theta'$ and $\theta''$ are Bayes optimal under realisability, it holds

$$\left| \sum_y \mathbb{P}_{\theta'}(\hat{Y} = y \mid Y = y) - \mathbb{P}_{\theta''}(\hat{Y} = y \mid Y = y) \right| = |\text{ACC}_{\theta'} - \text{ACC}_{\theta''}|$$

□

## E.2 FURTHER REMARKS ON THEORY ASSUMPTIONS AND BOUNDS

---

[2]Note that $\hat{Y}$ is parametrised by both $\eta$ and $\theta$.

### E.2.1 THEORY ASSUMPTIONS ON FINE-TUNING DATASETS

The assumption that $P(Y|A)$ is stable should always be satisfied if samples from $A$ are randomly selected when re-allocating the dataset. The assumption that $P(Y)$ is unchanged is more restrictive in practice. It is satisfied if both groups $A = 0$ and $A = 1$ have the same distribution of $Y$ (i.e., $P(Y|A = 0) = P(Y|A = 1)$). This is true for 10 out of 25 of the subgroups in our experiments (Table 2). While we relax this assumption for our experiments, and find the same consistent trend, we must make it for the purposes of the theorem, in particular so that we can isolate the effects of changes in A rather than changes in the label Y.

### E.2.2 CROSS-ATTRIBUTE EFFECTS

Theorem 5.1 extends naturally to scenarios where balancing with respect to another attribute $B$ indirectly alters the allocation of $A$, as discussed in Li et al. (2023). If the empirical distributions $P(A)$ and $P(Y \mid A)$ remain unchanged when reweighting by $B$, then the subgroup accuracies with respect to $A$ are stable. However, if $A$ and $B$ are correlated so that changing the allocation of $B$ induces a shift in the effective distribution of $A$, then the same bound applies with respect to the induced change in $A$. In particular, differences in subgroup accuracy across $A$ are bounded by the representation separation (TV distance) for $A$ and the magnitude of the change in $A$-allocation. Hence, balancing $B$ can only affect $A$-performance insofar as it implicitly changes the distribution of $A$ observed during fine-tuning.

### E.2.3 EXTENDING TO MULTIPLE DISCRETE GROUPS

We note that our Theorem 5.1 can also be extended to cases with non-binary attributes, i.e., for $A \in \{1, \ldots, K\}$. Define

$$\mathsf{TV}_K(\mathcal{D}) := \mathbb{E}_y \left[ \max_{a,b \in [K]} \mathsf{TV}\big( \mathbb{P}_\eta(Z \mid Y = y, A = a),\ \mathbb{P}_\eta(Z \mid Y = y, A = b) \big) \right].$$

Under the same assumptions, if $\mathsf{TV}_K(\mathcal{D}') \le \varepsilon$ and $\mathsf{TV}_K(\mathcal{D}'') \le \varepsilon$, then for all $a \in [K]$,

$$\left| \mathrm{ACC}_{\theta'}(A = a) - \mathrm{ACC}_{\theta''}(A = a) \right| \le 4\varepsilon + \left| \mathrm{ACC}_{\theta'} - \mathrm{ACC}_{\theta''} \right|.$$

### E.2.4 UPPER BOUND VALUES

In our experiments, we observe that $|\mathrm{ACC}_{\theta'} - \mathrm{ACC}_{\theta''}|$ is neglible. As shown in Figure 9, there is very little variation in overall model performance across fine-tuning allocations, with a mean accuracy standard deviation of 0.016, 0.002, 0.001, and 0.001 for MNIST, MIMIC, HAM, and Civil_comments respectively.

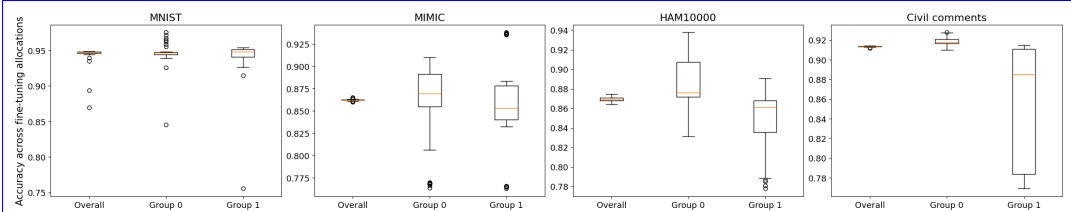

Figure 9: Overall and group-wise accuracy across all fine-tuning runs for MNIST, MIMIC, HAM10000, and Civil_comments.

We further calculate the value of the upper bound for each subgroup, and find that it is generally informative, with values between 0 and 0.4, except for very high TV subgroups, where it is sometimes

above 1. HAM10000 appears to be an outlier, as every subgroup (even random groups), have very high TV ($> 0.25$) causing the upper bound to be consistently above 1 (which the LHS trivially satisfies). This is most likely because the HAM10000 dataset is of a much smaller size (1000 images, with only 175 positive samples) which causes the TV estimates to be unreliable and very sensitive to small changes like the bin size used in histogram estimation.

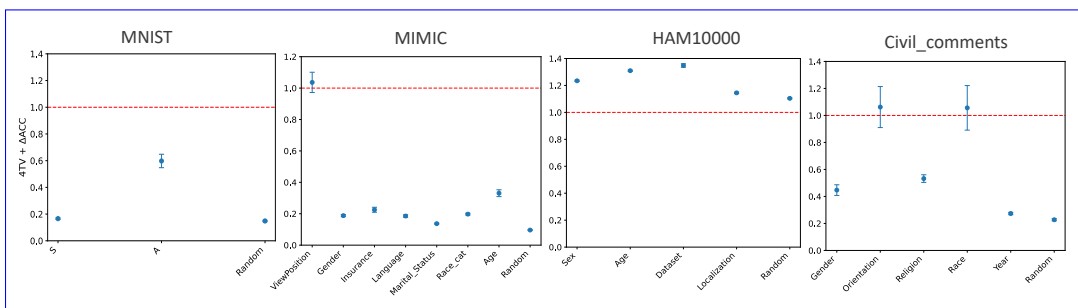

Figure 10: Value of upper bound for MNIST, MIMIC, HAM10000, and Civil_comments subgroups, with the trivial upper bound (accuracy difference of 1) marked in red.

## F   Supplementary results on representation invariance

### F.1   Additional details on distance metrics

We analyse the pre-trained models' representations by extracting the penultimate layer embeddings of the test inputs (e.g., 1024-dimensional vector $z$ for DenseNet121) and projecting them to a lower dimensional space via principal component analysis. To reduce noise, we retain the top-$k$ principal components that explain at least 70% of the variance (in practice $k \in [2, 81]$ depending on the dataset and model) and measure the mean **total variation distance** (TV) between the representations of examples in subgroups $a_0^{(j)}$ and $a_1^{(j)}$ in this lower dimensional space. We condition on $Y$ to control for class imbalances.

Given subgroup embeddings $z_{a_0^{(j)}}$ and $z_{a_1^{(j)}}$, we estimate their distributions by constructing normalized histograms along each principal component dimension. For a given component, let $p$ and $q$ denote the resulting discrete probability mass functions over bins $b$. The TV is then $\mathrm{TV}(p, q) = \frac{1}{2} \sum_b |p(b) - q(b)|$, where the sum is over histogram bins. In practice, we compute TV for each principal component dimension separately over 50 bins and report the average across dimensions. TV provides a bounded ($[0, 1]$) measure of separation. Values near 0 indicate nearly identical marginal distributions, while values near 1 indicate almost complete disjointness.

For completeness, we also explore additional distance metrics including the **Wasserstein distance** (WD) and the **Fréchet distance** (FD) which emphasise distinct representation differences.

For WD, we compute the 1D WD along each principal component vector of the embeddings and report the mean across components:

$$\mathrm{WD}\big(z_{a_0^{(j)}}, z_{a_1^{(j)}}\big) = \frac{1}{k} \sum_{i=1}^{k} W_1\big(z_{a_0^{(j)}}^{(i)}, z_{a_1^{(j)}}^{(i)}\big),$$

where $z_{a_0^{(j)}}^{(i)}$ and $z_{a_1^{(j)}}^{(i)}$ denote the projections of the embeddings from subgroups $a_0^{(j)}$ and $a_1^{(j)}$ onto the $i$-th principal component, and $W_1$ is the univariate Wasserstein-1 distance.

The FD approximates the subgroup feature distributions as multivariate Gaussians. Let $(\mu_{a_0^{(j)}}, \Sigma_{a_0^{(j)}})$ and $(\mu_{a_1^{(j)}}, \Sigma_{a_1^{(j)}})$ be the empirical means and covariances of the embeddings for the two subgroups. Then

$$\mathrm{FD}\big(z_{a_0^{(j)}}, z_{a_1^{(j)}}\big) = \|\mu_{a_0^{(j)}} - \mu_{a_1^{(j)}}\|_2^2 + \mathrm{Tr}\Big(\Sigma_{a_0^{(j)}} + \Sigma_{a_1^{(j)}} - 2(\Sigma_{a_0^{(j)}}^{1/2} \Sigma_{a_1^{(j)}} \Sigma_{a_0^{(j)}}^{1/2})^{1/2}\Big).$$

Together, these measures provide different estimates of on invariance: TVD emphasizes the largest discrepancies in probability mass between subgroups, WD gives a precise estimate of marginal distribution shift (including differences in distributional shape), and FD captures coarse differences in means and covariances.

### F.2   Representation distances obtained across the four datasets

We highlight the variation in TV across subgroups in the four datasets in Figure 11.

### F.3   Metrics are robust

All three distance metrics are correlated and appear robust to whether the calculation is done on the dimension reduced space or full feature space. We calculate representation distance on a dimension-reduced space to reduce noise (our test sample size is sometimes smaller than our latent embedding size) and for improved computational efficiency.

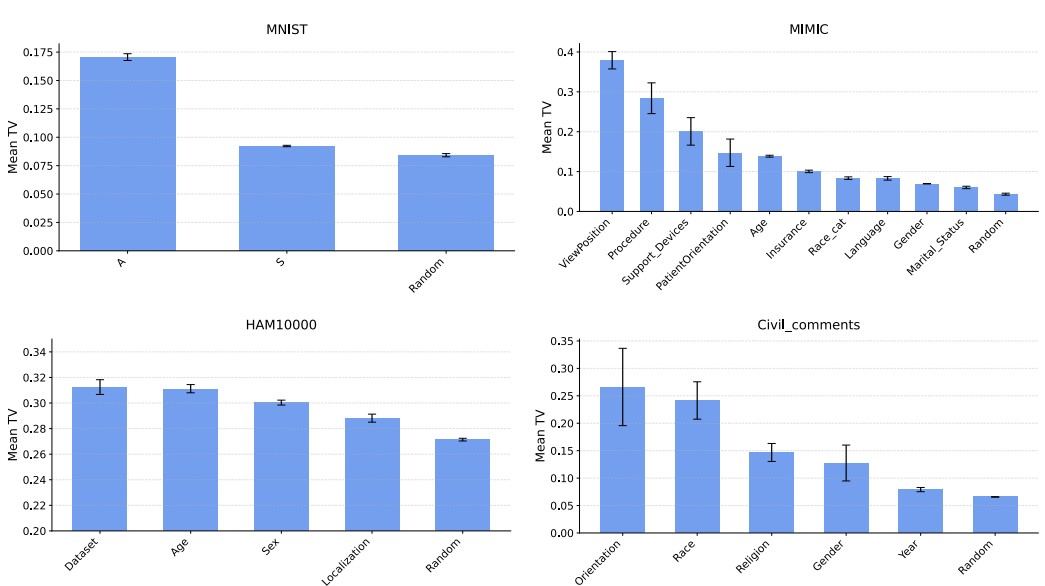

Figure 11: Total variation (TV) distances across subgroups in the pre-trained model's penultimate-layer representation space. We report mean and standard deviation across three random seeds.

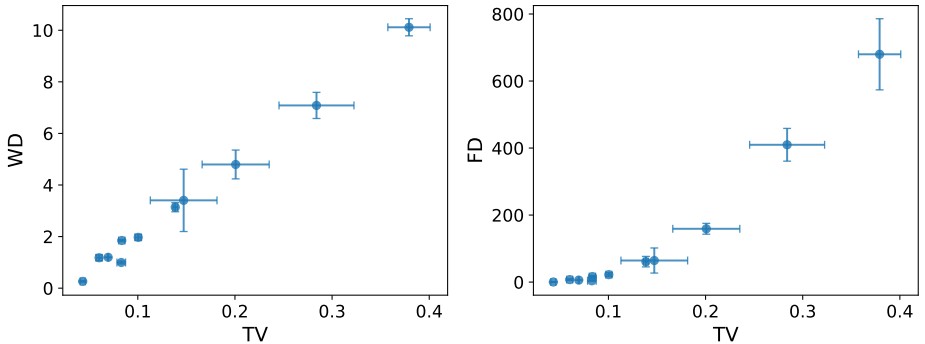

Figure 12: Total variation distance (TV), wasserstein distance (WD), and Fréchet distance (FD) across subgroups appear correlated in MIMIC. Each dot represents a subgroup with error bars representing standard deviation across three pre-training runs.

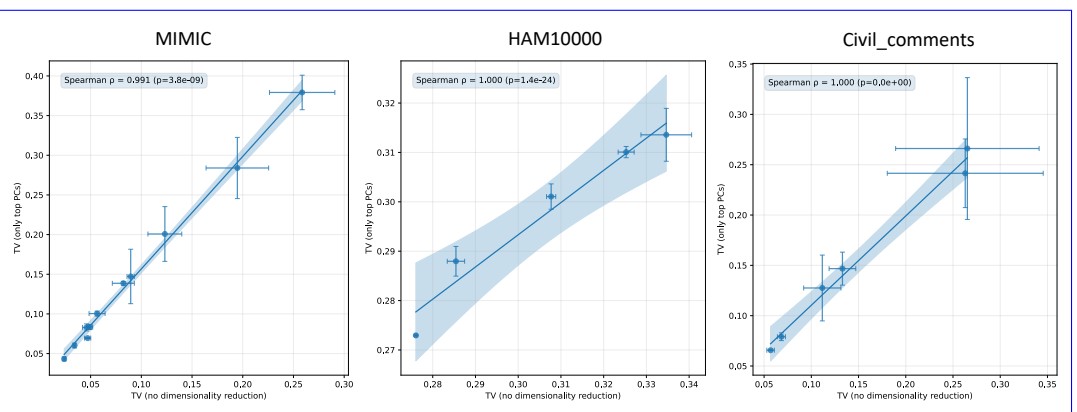

Figure 13: Representation distance metrics appear robust to whether they are calculated on the full latent space or on the lower dimensional space (after PCA). Each dot represents a subgroup in each of the three datasets with error bars representing standard deviation across three pre-training runs.

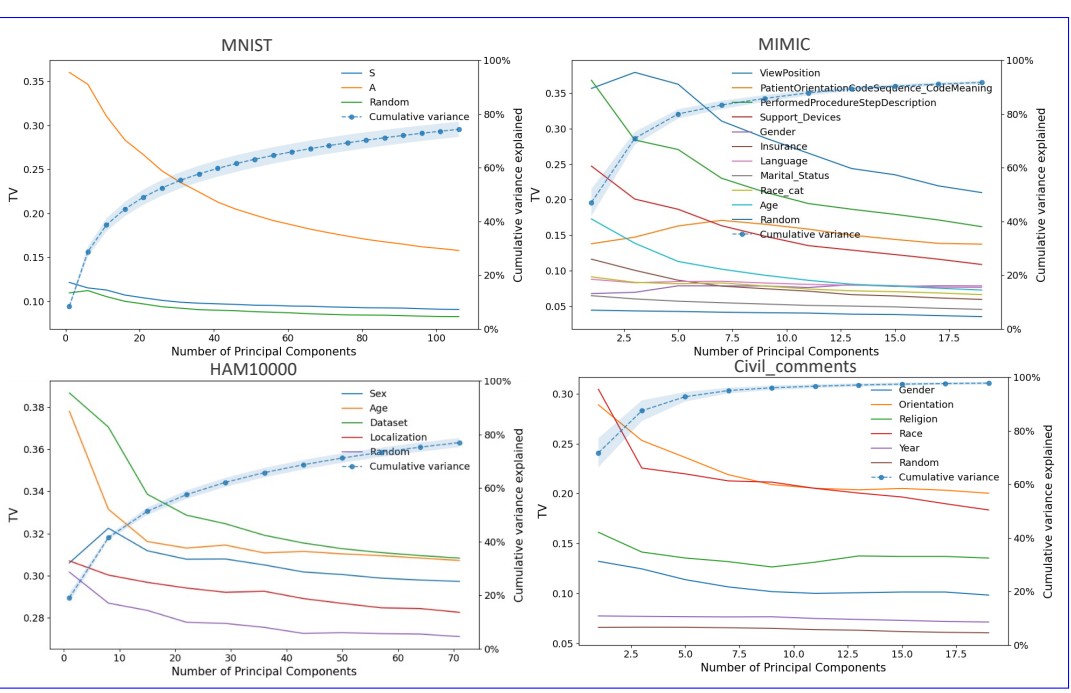

Figure 14: Representation distance metrics appear robust to the number of principal components on which they are calculated (for sufficiently high explained variance). Each line shows the estimated TV between one attribute calculated on an increasing number of principal component vectors.

## G   SUPPLEMENTARY RESULTS ON THE CORRELATION BETWEEN REPRESENTATION DISTANCE AND SUBGROUP ALLOCATION SENSITIVITY

Figure 15 extends the main text by showing subgroup allocation sensitivity against additional distance metrics across our datasets. The correlation is strong across metrics, supporting the latent separation hypothesis.

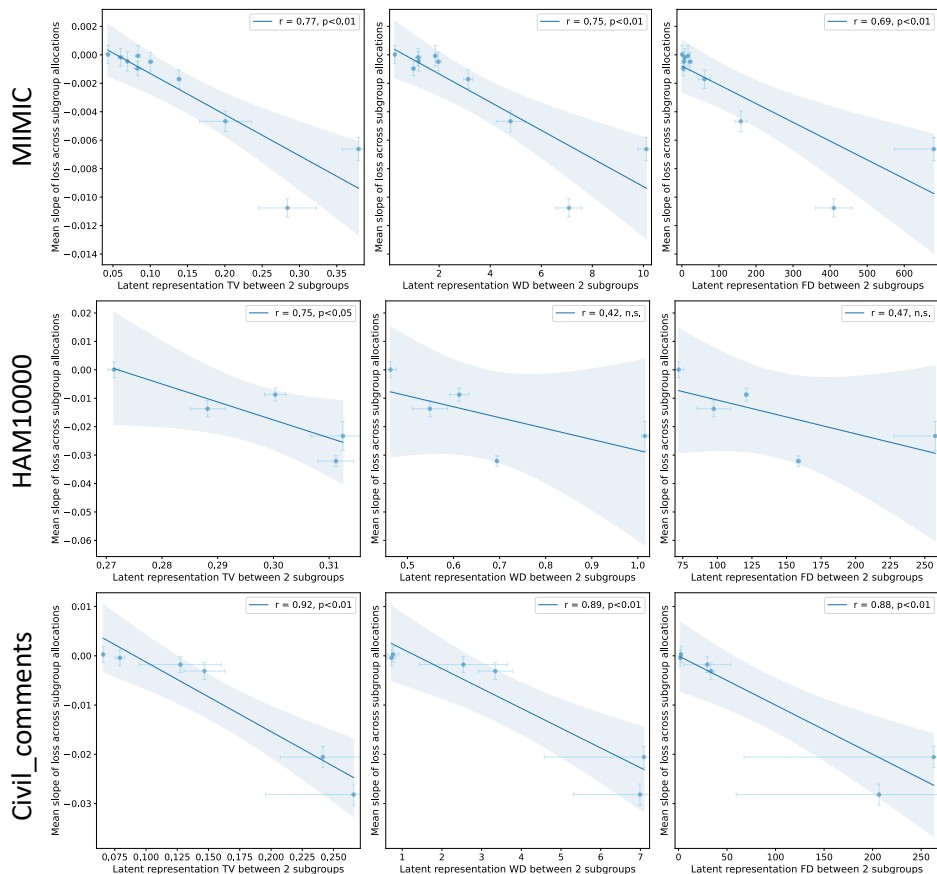

Figure 15: Sensitivity to subgroup allocation is correlated with separation in the pre-trained model's representation space (as measured by total variation distance, wasserstein distance, and Fréchet distance) across the three datasets. Each dot represents mean distance and loss slope for one subgroup, averaged across 9 fine-tuning runs, with bars corresponding to standard deviations, and Pearson correlation coefficients also shown.

# H EXTENDING SETUP TO OTHER TRAINING REGIMES

## H.1 FULL FINE-TUNING

We additionally extend our setup from last-layer fine-tuning to full model fine-tuning. Overall, the correlations are equally significant, and the magnitude of the effects larger (Figures 17 and 18). We attribute this to the fact that in practice, separation of subgroup representations does not change with subgroup allocations (Figure 16). Therefore, while some of our theoretical assumptions still hold, the effect increases as full fine-tuning allows for greater modification of network parameters.

We hypothesise that representation separation is so consistent between pre-training and fine-tuning and across allocations (Figure 16) because of non-spurious shifts in the features necessary to predict $Y$ across groups. For example, in MIMIC, the TV distance between different X-ray views (frontal or lateral) is consistently high. This is most likely because the necessary features to diagnose disease differ depending on the viewpoint, and therefore, across allocations, a model must maintain a high separation in its latent space to perform well on both groups.

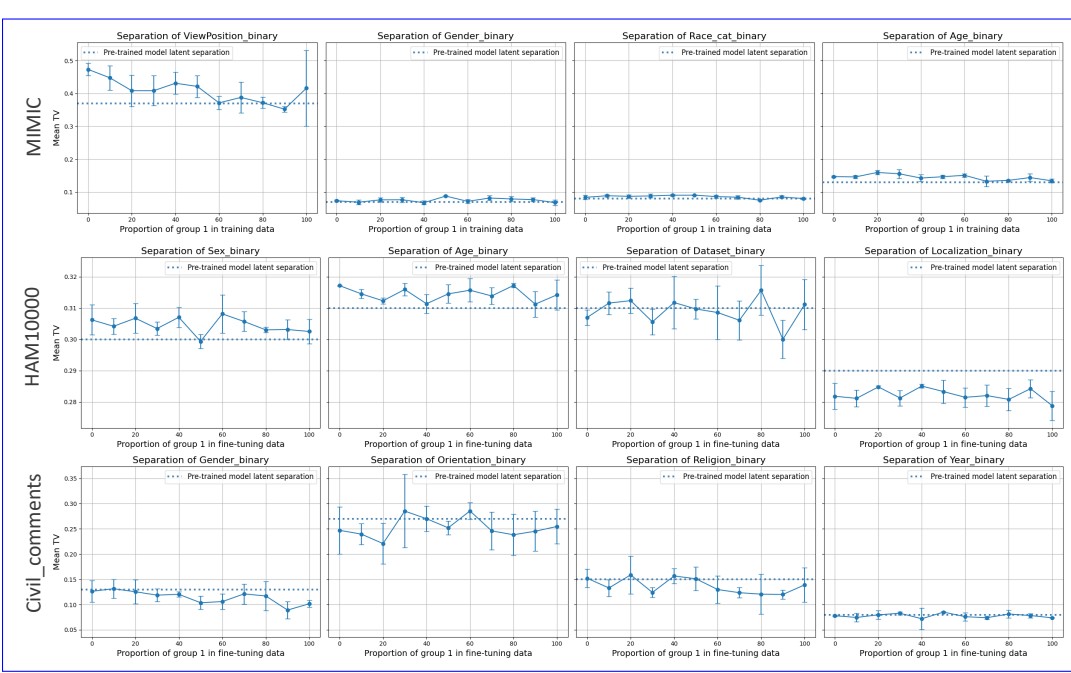

Figure 16: Examples showing latent representation separations are stable across subgroup allocations under full fine-tuning and close to separation of the initial pre-trained model.

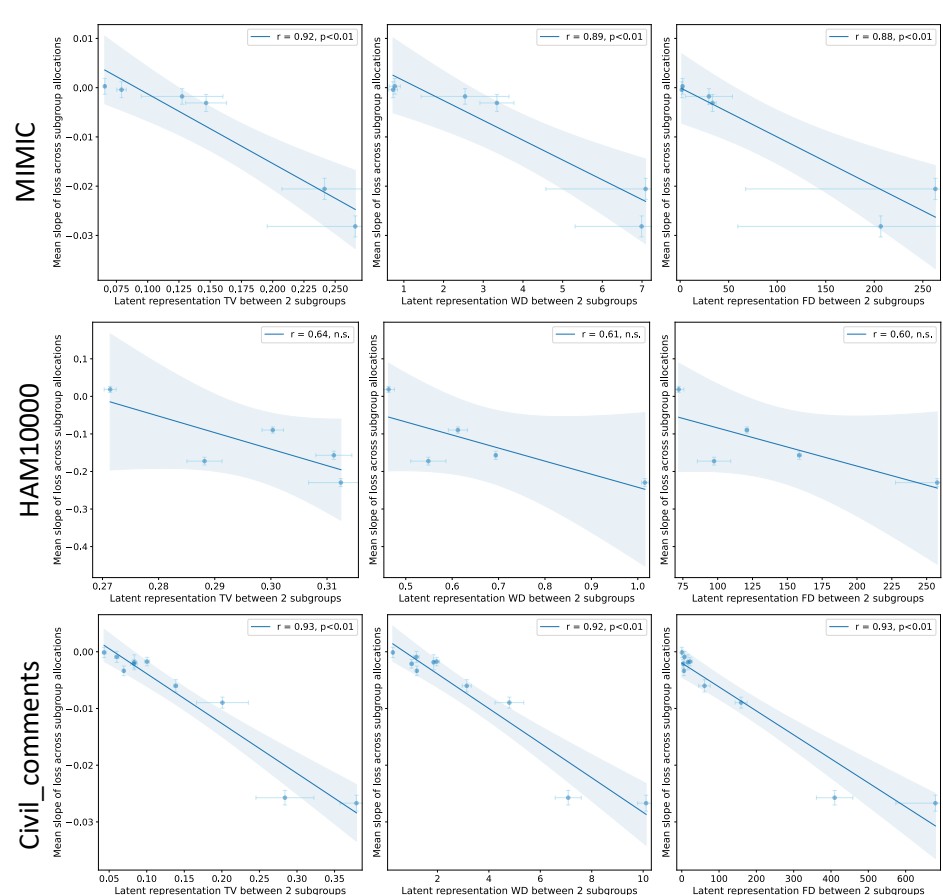

Figure 17: Correlation between loss slope and distance metrics remains strong under full fine-tuning. Each dot represents mean distance (total variation distance, wasserstein distance, or Fréchet distance) and loss slope for one subgroup, averaged across 9 fine-tuning runs, with bars corresponding to standard deviations, and Pearson correlation coefficients also shown.

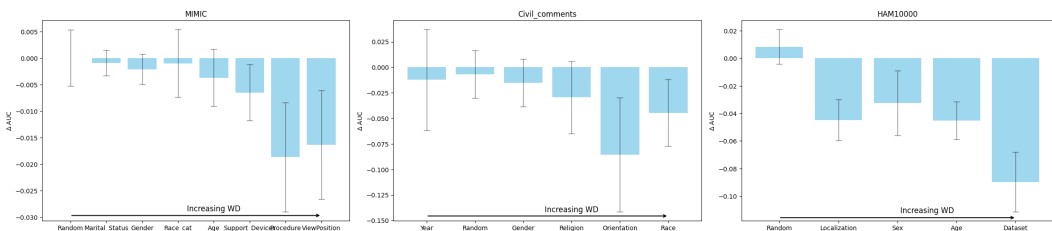

Figure 18: Representation distances continue to predict AUC gaps at extreme subgroup allocations under full fine-tuning at 100% allocation vs 0%. Bars represent mean and standard deviation of the AUC across 9 fine-tuning runs.

## H.2 TRAINING FROM SCRATCH

We also test whether the our hypothesis holds when only doing one round of training from scratch. We train the same models from scratch while systematically varying subgroup training data allocation and measure how much performance changes across allocations. We find similar trends in allocation sensitivity, but of a much greater magnitude (e.g., loss slopes are approximately 10x steeper than in last-layer fine-tuning), as expected since the models can vary more. We next explore whether this is linked to latent representation separation, using the model trained on the natural dataset proportions to extract latent embedding vectors. We find very similar results to fine-tuning, with a strong significant correlation between latent representation separation and sensitivity to subgroup allocation (Figures 19 and 20). While this finding is less clearly actionable under this setup (e.g., cannot directly inform subsequent data collection efforts), it provides a strong explanation as to the behaviour of models trained under different subgroup allocations.

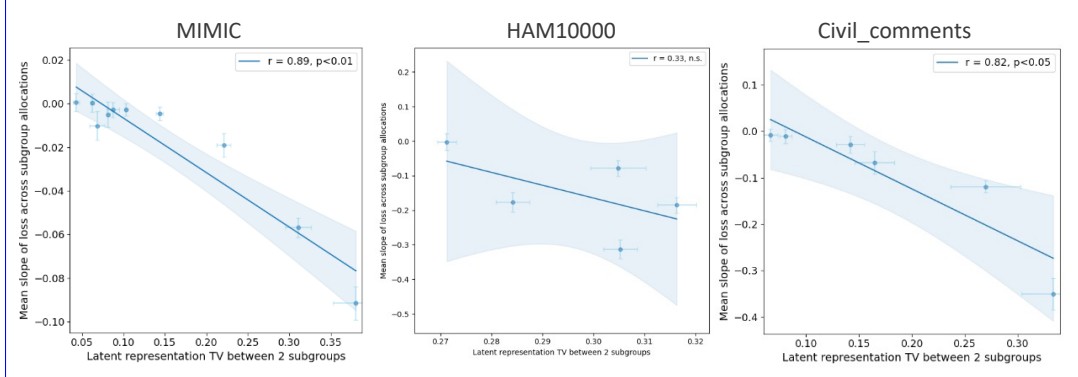

Figure 19: Correlation between loss slope and distance metrics remains strong when training from scratch. Each dot represents mean total variation distance and loss slope for one subgroup, averaged across 3 training runs, with bars corresponding to standard deviations, and Pearson correlation coefficients also shown.

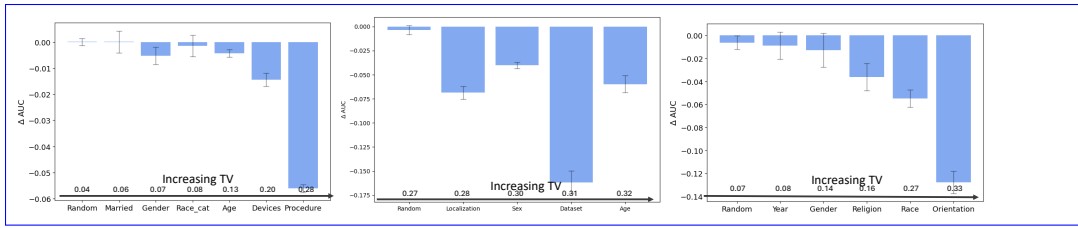

Figure 20: Representation distances continue to predict AUC gaps at extreme subgroup allocations when training from scratch at 100% allocation vs 0%. Bars represent mean and standard deviation of the AUC across 3 training runs.

# I  SUPPLEMENTARY RESULTS ON FOUNDATION MODEL FINE-TUNING

## I.1  SUPPLEMENTARY EXPERIMENTAL DETAILS

We test whether our results hold in the much less controlled setting of foundation model fine-tuning. We experiment with two vision-language models, CheXagent (Chen et al., 2024), and RAD-DINO-MAIRA-2 (Pérez-García et al., 2024). Both of these rely on fundamentally different training mechanisms. CheXagent is trained on images and text with SigLIP training while RAD-DINO-MAIRA-2 is only trained on images with a DINO-based setup, providing two distinct embedding regimes to assess the robustness of our hypothesis.

We first pass the MIMIC-CXR images through the image encoders `XraySigLIP_vit-l-16-siglip-384_webli` and `rad-dino-maira-2` and obtain 1024- and 768-dimensional embeddings respectively. We then train a single classification layer on each of the 16,000 MIMIC-CXR training embeddings, varying allocation in the same way as previous experiments. We test the classification model on the MIMIC-CXR test set (which, to the best of our understanding, neither model has been trained on).

## I.2  SUPPLEMENTARY RESULTS

We show differences in latent representation separations in both foundation models in Figure 21.

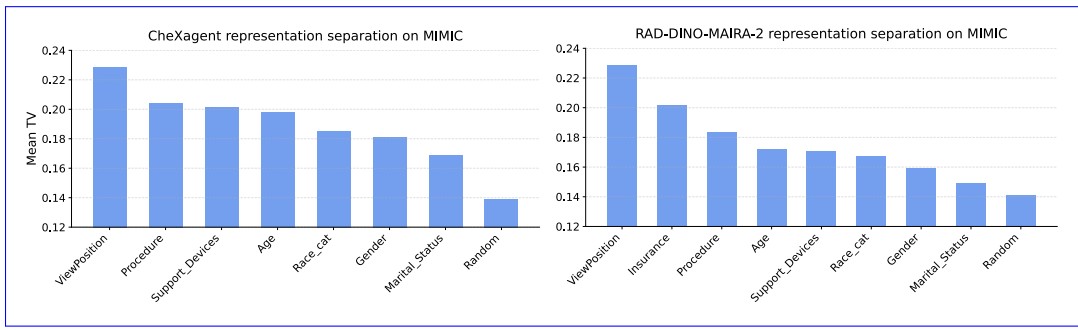

Figure 21: Total variation distances of CheXagent (left) and MAIRA-2 (right) embeddings of MIMIC images across subgroups.

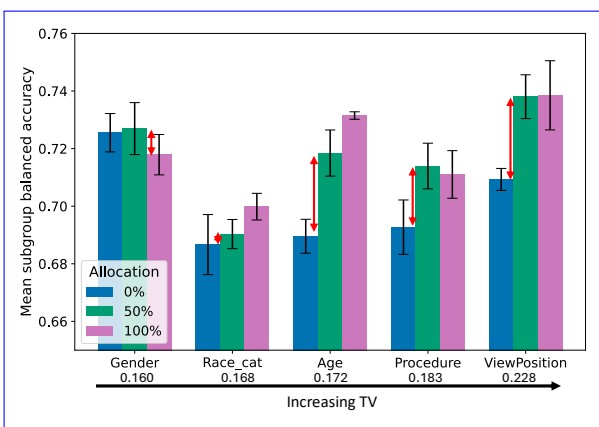

Figure 22: In RAD-DINO-MAIRA-2 foundation model fine-tuning, selecting a balanced allocation for imaging subgroups increases subgroup accuracy by over 0.03, while it has less importance for demographic subgroups, as predicted by their reduced total variation distance. We report mean accuracy and standard deviations for 3 fine-tuning runs, with red arrows for potential gains from balanced allocations.

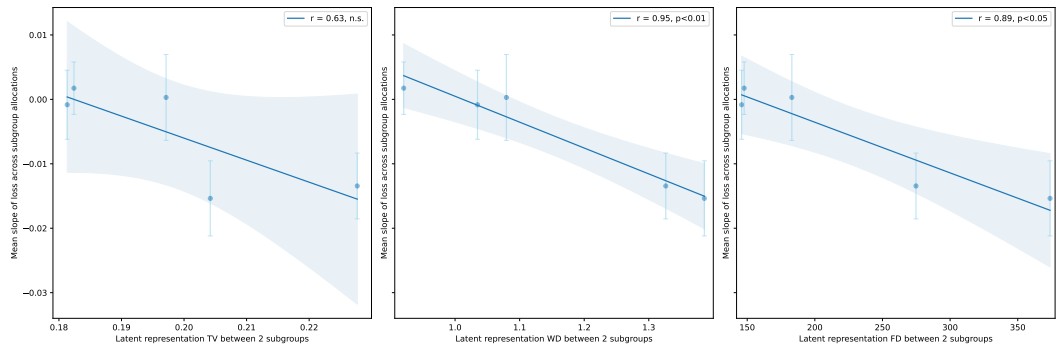

Figure 23: Correlation between latent representation separation and subgroup allocation sensitivity in CheXagent (vision-language model) fine-tuning. Results confirm that our hypothesis generalises beyond task-specific models. Each dot represents mean distance and loss slope for one subgroup, averaged across 9 fine-tuning runs, with bars corresponding to standard deviations, and Pearson correlation coefficients also shown.

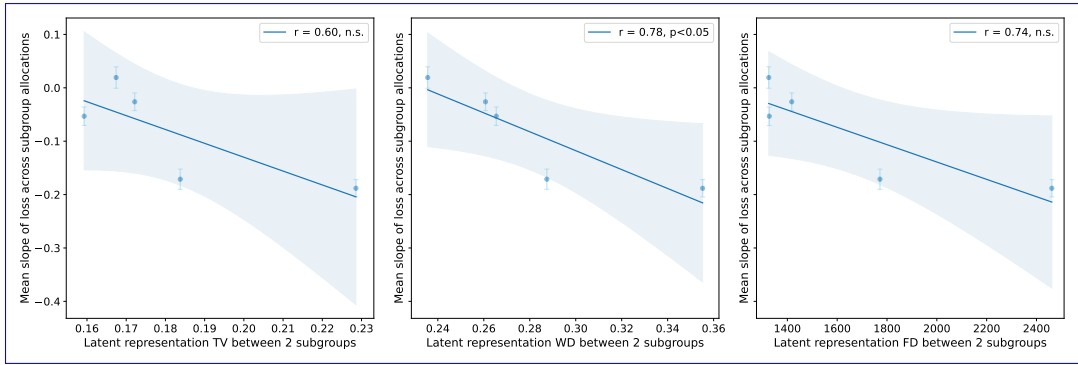

Figure 24: Correlation between latent representation separation and subgroup allocation sensitivity in MAIRA-2 (vision-language model) fine-tuning. Results confirm that our hypothesis generalises beyond task-specific models. Each dot represents mean distance and loss slope for one subgroup, averaged across 9 fine-tuning runs, with bars corresponding to standard deviations, and Pearson correlation coefficients also shown.

# J  ABLATIONS ON FINE-TUNING BUDGET $K$

We repeat the MIMIC experiments with a smaller fine-tuning budget $K$ to explore whether changing sample size modifies our results. We find that smaller budgets show slightly higher absolute magnitudes of allocation sensitivity. This is most likely due to larger sample sizes allowing the model to learn more robust disease representations which generalise better across allocations. However, correlations are equally significant across fine-tuning budgets, suggesting our latent representation hypothesis is valid across dataset sizes.

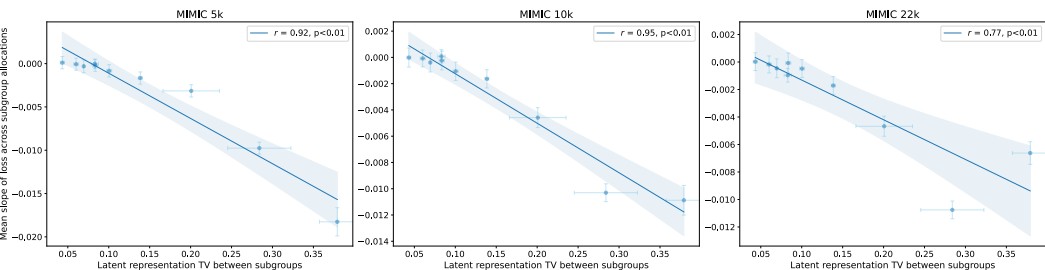

Figure 25: Correlation between representation separation and subgroup allocation sensitivity is strong across fine-tuning dataset sizes in MIMIC. Each dot represents mean distance and loss slope for one subgroup, averaged across 9 fine-tuning runs, with bars corresponding to standard deviations, and Pearson correlation coefficients also shown.

## K   IMPLEMENTATION OF TV-BASED LATENT REGULARISATION

During pre-training, we augment the standard cross-entropy loss with a differentiable surrogate for the TV distance between the embedding distributions of the two view-position groups. For each mini-batch, we compute a Mahalanobis-squared distance between the group-wise mean last-layer embeddings within each label (i.e., conditioned on $Y$), and average these distances. This quantity acts as a smooth proxy for the TV divergence and is scaled by a regularisation hyperparameter $\lambda$. The resulting loss is

$$\mathcal{L} = \mathcal{L}_{\text{task}} + \lambda \, \widehat{\text{TV}}(Z \mid A, Y),$$

where $A$ denotes the view-position group and $Z$ the last-layer embeddings. No gradient is taken through density estimates; the surrogate is computed directly from batch statistics and therefore adds minimal computational overhead. In practice, we use $\lambda = 0.01$, as this gives approximately equal weighting to the CE objective and the distance estimate (which is on the order of 100). To evaluate sensitivity to subgroup allocation, we fine-tune this model at with different view proportions as in our standard pipeline.

## L  LLM USAGE

We used a large language model (OpenAI ChatGPT) as a general-purpose assistant tool to help with phrasing and grammar improvements. The model was not used for research ideation, experiment design, data analysis, or generation of original scientific content. All technical ideas, methods, and results are the authors' own, and the authors take full responsibility for the paper's contents.

