# OpenReview forum: "Representation Invariance and Allocation: When Subgroup Balance Matters"
_ICLR.cc/2026/Conference — Submitted to ICLR 2026_

### Official Review · Reviewer_zYwC · 2025-10-23

**Soundness:** 3
**Presentation:** 3
**Contribution:** 2
**Rating:** 4
**Confidence:** 4

**Summary:**

This paper investigates why subgroup accuracy is sometimes drastically affected by data allocation during finetuning, while other times entirely unaffected even by missing an entire subgroup. A property of the learned representation, called the latent separation and defined by the total variation distance between the embedding distribution for data with and without the spurious attribute, is proposed as an explanation. The latent separation and subgroup sensitivity are theoretically linked, and experiments are provided which underscore their strong empirical correlation.

**Strengths:**

1. The correlation studies in Section 4.2 and 6.3 are exceptionally well-done. The variety of datasets, discussion on least-squares vs power-law fit, exploration of different latent distance metrics, and use of metrics beyond the standard worst-group accuracy all contribute to an informative and precise investigation. Overall, a strong commitment to scientific rigor is obvious.

2. The latent separation hypothesis makes intuitive sense and is justified both theoretically and empirically. The experiments in Section 6.3 and 6.4 show that latent separation is indeed well-correlated with subgroup allocation sensitivity, even in a realistic foundation model finetuning setting. The use of the total variation distance between the embedding distributions is, to my knowledge, quite novel.

**Weaknesses:**

The main weakness of the paper is the limited scope of its findings.

1. The proposed latent separation hypothesis is not strong enough to answer the key questions presented in the Introduction or the insufficiency of current explanations (such as that in Section 4.3). A good example is the phenomenon that “imbalanced data distributions actually improve subgroup performance” referenced several times in the paper. The proposed theory only explains when certain groups are sensitive to mis-allocation (i.e., when the latent separation is large), and _not_ why the optimal allocation for a known sensitive subgroup may not be a uniform allocation. I am not asking for a method which _predicts_ optimal allocation, as the authors address in Section 6.5, but instead some insight relating subgroup representations and allocation which is more fine-grained than a relative ordering of sensitive subgroups (c.f. Sections 6.3 and 6.4).

2. The operationalization of the proposed latent separation hypothesis is somewhat incomplete, and its findings could be further validated by algorithms leveraging its insights. Here are two ideas:

    a. The latent separation metric could be used as an interpretability device for discovering the most sensitive subgroups within a dataset, which likely correspond to those with substantial robustness concerns. Specifically, given a dataset with unknown group annotations, compute the partition of the data such that the TVD of the embedding distribution between subgroups is maximized. This could help users discover previously unknown biases (an active area of interpretability research, e.g., [1]) and it would validate the latent separation hypothesis if this partition corresponds to a priori known spurious correlations.

    b. The latent separation metric could also be used as a regularizer during training to mitigate sensitivity to certain attributes and encourage the model to learn more generalizable and invariant representations. If regularizing the TVD of known sensitive subgroups leads to a reduction in sensitivity, it would likewise validate the latent separation hypothesis.

3. The theory serves its purpose in connecting the TVD of the embedding distributions to subgroup sensitivity in a simple linear setting. However, it is not too surprising, as it essentially states that if the representation is invariant to a particular attribute, then the proportion of data with that attribute doesn’t matter too much (which follows intuitively from the invariance property). More importantly, similar to point (1), it is not strong enough to provide insight into the key questions in this space. In particular, the question of what training mechanism induces different levels of latent separation for different subgroups is entirely avoided, yet would contribute substantially to empirical and theoretical lines of work in this direction [2, 3, 4].

[1] Moayeri et al. Spuriosity Rankings: Sorting Data to Measure and Mitigate Biases. NeurIPS 2023.

[2] Izmailov et al. On Feature Learning in the Presence of Spurious Correlations. NeurIPS 2022.

[3] Yang et al. Identifying Spurious Biases Early in Training through the Lens of Simplicity Bias. AISTATS 2024.

[4] Qiu et al. Complexity Matters: Dynamics of Feature Learning in the Presence of Spurious Correlations. ICML 2024.

**Questions:**

1. How is this work contextualized in the broader line of research at the intersection of representation learning and spurious correlations/group robustness? There is a substantial body of work investigating properties of the representation to understand subgroup robustness: see, e.g., [1, 2, 3, 4, 5].

2. Two additional references for the point that perfect subgroup balance may not achieve optimal accuracy which should be discussed: [6, 7]

3. In Section 5, the notation $\mathbb{P}\_\theta$ is a bit confusing. Why is the distribution of $Z$ parameterized by model $\theta$ when it is only $h\_\eta$ which induces $Z$, independent of $g_\theta$? It should also be made clear that $\mathbb{P}\_\theta$ is a distribution/measure and not a probability. Lastly, is it important that both TVDs are bounded by the same $\epsilon$, or does the $4\epsilon$ generalize to $2\epsilon\delta$ if one TVD is bounded by $\epsilon$ and the other by $\delta$?

4. Minor clarity/grammatical improvements:

    a. Missing colon line 83

    b. “Following” misspelled line 268

[1] Izmailov et al. On Feature Learning in the Presence of Spurious Correlations. NeurIPS 2022.

[2] Bahng et al. Learning De-biased Representations with Biased Representations. ICML 2020.

[3] Arjovsky et al. Invariant Risk Minimization. ArXiv 2019.

[4] Zhang et al. Rich Feature Construction for the Optimization-Generalization Dilemma. ICML 2022.

[5] Hermann et al. On the Foundations of Shortcut Learning. ICLR 2024.

[6] Qiao et al. Group-robust Sample Reweighting for Subpopulation Shifts via Influence Functions. ICLR 2025.

[7] Schwartz et al. On the Limitations of Dataset Balancing: The Lost Battle Against Spurious Correlations. NAACL 2022.

---

> ### Author Response · Authors · 2025-11-18
> **Response to Reviewer Comment**
>
> We thank the reviewer for their insightful comments and the depth of review they have provided. We are pleased to hear they found our “correlation studies [...] are exceptionally well done” and our hypothesis intuitive and well-justified. We respond to each comment point-by-point below, with references to changes we have made in the revised manuscript (which we have re-uploaded).
>
> > W1: Latent separation hypothesis is not enough to answer key questions.
>
> We agree with the reviewer that we do not explain what the optimal allocation is when a subgroup has **low TV**. All we say is that when subgroups are not significantly separated in the model’s latent space, **their performance *should not* depend on their allocation**. Beyond this, we cannot make a statement on whether there is a particular non-uniform allocation which will be optimal because we do not have the theoretical tools to model other factors which may affect this (e.g., noise in certain samples, certain samples being more or less “informative”, confounders etc.). While this would definitely be interesting future work, at this stage this seems like an unrealistic research question.
>
> Furthermore, for cases with **high TV**, if the practitioner cares equally about maximising performance across both groups (say, men and women), then a **balanced allocation is optimal**. Again, this is assuming all male and female samples are equally informative and noisy etc., which may not be true in practice.
>
> Not only do we believe these statements are **as precise as we can make them given the inherent limitations to modelling real-world data**, but we also believe they are **as precise as would really make sense for a real-world use-case** (as we mention in Section 6.5). Our hypothesis is already much more informative than any existing work, which either a) requires dozens of models to be trained [1] or b) has ungrounded intuitions which do not necessarily hold true [2,3].
>
> We thank the reviewer for highlighting this point, and have slightly **lengthened our discussion on it in section 6.5** of the revised paper.
>
> > W2: Operationalisation of the hypothesis is incomplete, ideas could include a) use as an interpretability method or b) use as a regulariser during training.
>
> We appreciate the reviewers’ suggestions and think they could definitely strengthen the paper. **We are completing additional experiments** and will get back to the reviewer when they are finished.
>
> > W3: What training mechanism induces different levels of latent separation for different subgroups?
>
> While this is definitely a valuable research question, a) it has already been the **focus of other studies** [4-7] (as the reviewer mentions), and b), it **does not completely align with the positioning of our paper.** Our main aim is to show that, *given* a model (and its representations), subgroup allocation sensitivity can be variable, and to explain and justify why this is the case. Although our hypothesis may not be surprising, we are the first to formalise and justify it. We are not trying to modify the state of the model (e.g., by proposing a new bias mitigation method), nor are we arguing that subgroup latent separation should always be avoided [4]. Given our focus, a detailed study of training mechanisms that induce different levels of separation would be orthogonal to our contribution. We have now added a note about this to our future work section. We are happy to discuss this point further and reconsider if the reviewer still believes this would be a valuable addition.
>
> [1] Rolf et al., Representation Matters: Assessing the Importance of Subgroup Allocations in Training Data, ICML 2021.
>
> [2] Idrissi et al, Simple data balancing achieves competitive worst-group-accuracy, PMLR 2022.
>
> [3] Kirichenko et al., Last Layer Re-Training is Sufficient for Robustness to Spurious Correlations, ICLR 2023.
>
> [4] Izmailov et al. On Feature Learning in the Presence of Spurious Correlations. NeurIPS 2022.
>
> [5] Hermann et al. On the Foundations of Shortcut Learning. ICLR 2024.
>
> [6] Yang et al. Identifying Spurious Biases Early in Training through the Lens of Simplicity Bias. AISTATS 2024.
>
> [7] Qiu et al. Complexity Matters: Dynamics of Feature Learning in the Presence of Spurious Correlations. ICML 2024.
>
> [8] Jones et al, A causal perspective on dataset bias in machine learning for medical imaging, Nature Machine Intelligence 2024.

---

> > ### Author Response · Authors · 2025-11-18
> >
> > > Q1: How is this work contextualized in the broader line of research at the intersection of representation learning and spurious correlations/group robustness?
> >
> > We thank the reviewer for pointing out these references. These works all broadly **share the same premise that invariant or robust representations lead to better generalisation** (e.g., across groups or across non-spuriously correlated samples). In particular, the core theory introduced in Invariant Risk Minimization [1] matches our theory and experiments as we find that invariant representations lead to the same model generalising to groups (environments) it has not been trained on (or has been trained very little on). Our work extends these initial theories to the fairness literature, and crucially, shows how this invariance is related to subgroup data balancing behaviours. We had already mentioned this in 6.3 but have now **elaborated on it a bit further and mentioned additional works** in fair representation learning.
> >
> > With respect to the works on spurious correlations (SC) and representation learning [2-4], our work differs in two main ways. Firstly, those papers analyse or learn robust representations (e.g., via invariance penalties), but do not provide any connection to how training data allocation affects subgroup performance. Secondly, spurious correlations are only one example of the groups that could be relevant in fairness work, and are much simpler to study than, for instance, real demographic groups. For example, in SCs, there is often a clear distinction between “core features” and “spurious features” [2] which is very much not the case for most demographic groups. We choose to focus on a broader set of groups as these are less well studied and characterised.
> >
> > Overall, our work **differs fundamentally from past works on robust representation learning in the question we try to answer: “When will subgroup data balancing improve subgroup performance?”**, not “How can we learn invariant representations?”.
> >
> > > Q2: Additional references on subgroup balance not being optimal.
> >
> > We thank the reviewer for these pertinent references, we have now **added them to our related work section**.
> >
> > > Q3: Confusion theory notation and TV bound.
> >
> > We thank the reviewer for highlighting this, we understand the confusion. The reviewer is correct that $Z$ is only parametrised by $h_{\eta}$. We have **modified the notation** in our revised paper. We have also clarified that $P_{\eta}$ is a distribution. And finally, yes the reviewer is absolutely correct to note that if two TVs are bounded by different constants ($\varepsilon$ and $\delta$), then the upper bound can be reduced to: $2\varepsilon + 2\delta + | Acc_{\theta'} - Acc_{\theta''} |$. We have added a **note indicating this in the appendix**.
> >
> > > Q4: Minor clarity improvements.
> >
> > Edited now, thank you.
> >
> > We hope that our additional clarifications, manuscript revisions, and experiments will give the reviewer confidence to raise their score, and we welcome any further questions or feedback.
> >
> >
> > [1] Arjovsky et al. Invariant Risk Minimization. ArXiv 2019.
> >
> > [2] Izmailov et al. On Feature Learning in the Presence of Spurious Correlations. NeurIPS 2022.
> >
> > [3] Bahng et al. Learning De-biased Representations with Biased Representations. ICML 2020.
> >
> > [4] Hermann et al. On the Foundations of Shortcut Learning. ICLR 2024.

---

> > > ### Author Response · Authors · 2025-11-25
> > > **Response to Reviewer Comment (follow-up)**
> > >
> > > > W2b: "The latent separation metric could also be used as a regularizer during training to mitigate sensitivity to certain attributes and encourage the model to learn more generalizable and invariant representations. If regularizing the TVD of known sensitive subgroups leads to a reduction in sensitivity, it would likewise validate the latent separation hypothesis."
> > >
> > > We are following up with additional results related to the reviewer's suggestion on the operationalisation of the our hypothesis. As the reviewer suggested, we decided to intervene directly on the latent separation to test whether this leads to a reduction in sensitivity to subgroup allocation, as our hypothesis would predict.
> > >
> > > To do this, we **trained our MIMIC model with an added regularisation term to penalise last-layer embedding separation** between frontal and lateral X-rays (conditioned on the $Y$). Direct TV calculation is non-differentiable and computationally expensive, so instead we added a proxy term based on the squared Mahalanobis distance between the embeddings of frontal and lateral X-rays. We found that this indeed led to a reduced TV separation between frontal and lateral X-ray representations (70% lower than in the baseline model). Next, we explored whether the regularised model would be less sensitive to different fine-tuning frontal/lateral allocations and found that indeed, while the unregularised model's subgroup accuracy increased at a rate of 0.016 with increased allocation, the regularised model showed no increase at all (the accuracy slope was actually slightly negative, at -0.007).
> > >
> > > The results from this intervention strengthen our hypothesis, suggesting that **not only is latent TV separation associated with sensitivity to subgroup allocation, but that it is a key mechanism** behind it. **We have added these results to the main body of our revised manuscript (Figure 6)**.
> > >
> > > We thank the reviewer for this suggestion and hope that these new results, in addition to the previous explanations and manuscript updates detailed above, will give the reviewer confidence to raise their score.

---

> > > > ### Comment · Reviewer_zYwC · 2025-11-26
> > > >
> > > > I appreciate the authors' detailed response. I am especially interested in the follow-up experiments on the TV regularization idea: while average accuracy decreases by 10% (somewhat expected), the sensitivity to subgroup allocation decreases. It would be great to include a more comprehensive analysis in the final version of the paper.
> > > >
> > > > Overall, this paper presents insights that are relevant and interesting to the ICLR community; while limited in scope (as also noticed by Reviewer zhu1), the operationalization of the TV regularization is a step in the right direction. I have increased my score accordingly.

---

> > > > > ### Author Response · Authors · 2025-11-26
> > > > >
> > > > > We thank the reviewer for their engagement, and are pleased to hear our responses and extra experiments led them to increase their score. We agree that the TV regularisation idea strengthens our main claims, and will include a more comprehensive analysis in the final version of the paper. Thanks again for the feedback and constructive comments.

---

### Official Review · Reviewer_zhu1 · 2025-10-29

**Soundness:** 3
**Presentation:** 3
**Contribution:** 2
**Rating:** 4
**Confidence:** 3

**Summary:**

This paper presents a novel and comprehensive study on the effect of latent representation allocation in pre-trained models on the subgroup performance on the later fine-tuning. Both rich experiments and detailed theoretical analysis demonstrate that the sensitivity to subgroup allocation is highly correlated with the difference between the latent representations of two subgroups in the pre-trained model. The findings and discussions in this paper also help explain previously controversial observations regarding the relationship between subgroup performance and their group ratios and re-weighted loss.

**Strengths:**

(S1) Subgroup/fairness-related problems have been a very important topic in recent years, and this paper investigated the underlying representations of the subgroups to explain model performance instead of focusing on the attributes of the training dataset. I believe the research directions discussed in this paper are very promising and can complement other directions for a better understanding of the problems. Also, it aligns well with the interests of ICLR and positions the paper well among literature.

(S2) Detailed analysis and rich experiments clearly demonstrate that the representations from the pre-trained model have a big impact on later subgroup performance after the fine-tuning stage.

(S3) In general, the paper is well written and easy to follow. The problem setup and methods are well described in formal and rigorous language and equations.

**Weaknesses:**

(W1) The major weakness of this paper is that the scope of the problems and contributions is limited. While the authors discussed the general motivation of understanding how subgroup allocation affects subgroup performance in the introduction, the later problem setup is limited to a pre-training–then–fine-tuning setting. While some existing subgroup works used pre-trained models (e.g, Weng et al, 2023), others did not (e.g, Čevora et al, 2025), or were not specified. In fact, I don’t see that subgroup/fairness works have to rely on pre-trained–fine-tuning settings. As a result, although the key findings help explain some phenomena in those pre-trained-model-based works, in general, I believe readers won’t be surprised by the key findings of this paper that the representations from pre-trained models have a big effect on later fine-tuning subgroup performance.

(W2) In extension to W1, and more importantly even within this paper’s scope, the important question of how subgroup allocation affects subgroup performance remains unclear without comprehensive evaluations of the changes and patterns of the representations learned before and after fine-tuning. In other words, while we know that the difference of representations in pre-trained models has a big impact, how do the representations change after fine-tuning? Do they change at all? Why can’t fine-tuning correct the biased representations from pre-trained models, even with a larger ratio for the minor groups? What are the patterns of representation changes under different group ratios? The authors are encouraged to conduct comprehensive evaluations of the representations before and after fine-tuning and analyze their changes' relationship with subgroup allocation. Otherwise, the current findings, which mainly focus on the pre-trained model’s representations, are limited.

**Questions:**

(Q1) The authors are encouraged to respond to my concerns in the Weaknesses section, especially W2. Additional experimental results or simple demonstrations would be greatly appreciated and will influence my future decisions.

---

> ### Author Response · Authors · 2025-11-18
> **Response to Reviewer Comment**
>
> We appreciate the reviewers’ insightful comments and suggestions, and are glad to hear they find the research directions “very promising”. We agree with both points the reviewer has raised and have made changes to the manuscript to address them, which we have re-uploaded. We hope these additional experiments and clarifications will give the reviewer confidence to raise their score, and we remain open to discussing further.
>
> >W1: Scope limited to pre-train/fine-tune setting.
>
> We thank the reviewer for highlighting this, it is a good point. We completely agree that the scope of subgroup/fairness studies is not limited to the pre-train/fine-tune setting. We focused on this setup because we wanted to be able to provide actionable guidance, i.e., if a practitioner has a model they want to improve, they can first evaluate the latent representations of their base model, and then use our findings to inform what type of data they should use for fine-tuning. In the traditional single-round-of-training setup, our findings are less prescriptive.
>
> However, our initial explorations had actually focused on this setup. We trained many models from scratch across different training data allocations to see whether we could explain variation in subgroup balancing behaviours. Here, we also found that the latent representation separation of attributes (as measured on the model trained on the original dataset proportions) were extremely correlated with the models’ sensitivity to subgroup allocation. This finding provides an interpretability insight into why we observe different subgroup balancing behaviours even when training from scratch.
>
> We recognise that other readers may also be interested in this point, and have added these results to our Appendix H2. **Figures 18 and 19 summarise the trends we observe across the three datasets and demonstrate that the same correlations hold, if not more strikingly, as in the pre-train/fine-tune setting.** We thank the reviewer for noting this, and hope it helps convince them that our latent representation hypothesis is a more general phenomenon.
>
> >W2: Evaluation of the representations learned before and after fine-tuning.
>
> We completely agree that this is an important point to explore and thank the reviewer for this suggestion. We found that the **representations actually change very little after full fine-tuning**, at least as measured by the TV distance between attributes. We have added **Figure 15 to our Appendix H**, which shows that the TV distance across attributes is remarkably consistent in the three datasets after full fine-tuning at different fine-tuning allocations, with the variation across subgroup allocations rarely exceeding the standard deviation across runs. Furthermore, the separation is very close to the separation in the initial pre-trained model’s latent space (as shown by the blue dotted line in Figure 15). This stability helps explain why the latent separation observed in the pre-trained model remains highly predictive of sensitivity, even when the full network is fine-tuned.
>
> With respect to the question “Why can’t fine tuning correct the biased representations from pre-trained models even with a larger ratio for the minor groups?”, we would first like to highlight that we do not necessarily think a “biased representation” (i.e., a highly separated one) is inherently negative [1]. For example, in MIMIC, we find that the highest TV distance is between different views of x-rays (frontal or lateral). The necessary features to diagnose a disease ($Y$) likely differ depending on the viewpoint, meaning a model must maintain a high separation in its latent space to perform well on both groups.
>
> Therefore, in cases where the label $Y$ has different causal features across groups, it makes sense that even at extreme allocations the latent separation stays constant. Conversely, if the separation were due to a purely spurious correlation (e.g., $A=1$ is spuriously correlated to $Y=1$), then training only on $A=0$ should reduce the latent separation between $A=0$ and $A=1$, as the model should be able to use only the causal features of $Y$ to predict across both groups. In our work we believe all our subgroups pertain to the former case, which explains why if the representations are initially biased, they remain biased after fine-tuning.
>
> We appreciate the reviewer highlighting this interesting point, and **we have also further elaborated on this in Appendix H1**.
>
> [1] Jones et al, A causal perspective on dataset bias in machine learning for medical imaging, Nature Machine Intelligence 2024.

---

> > ### Author Response · Authors · 2025-11-27
> > **Follow up**
> >
> > We thank the reviewer again for their initial thoughtful review. We believe the additional results showing the same trends in the training-from-scratch setting and our extra analysis and discussion on representation changes before and after fine-tuning have further strengthened our hypothesis. We are grateful to the reviewer for suggesting this. We hope this has given the reviewer confidence to raise their score, and are happy to further engage if there are any final points we can clarify.

---

### Official Review · Reviewer_5CBB · 2025-10-30

**Soundness:** 3
**Presentation:** 2
**Contribution:** 3
**Rating:** 4
**Confidence:** 4

**Summary:**

Past work has proposed models that look at how the allocation of (pre)training data across subgroups influences ML model performance on different subgroups -- often making an intuitive assumption that adding data from certain groups will not decrease performance on those groups (and indeed, will generally increase it). However, ascertaining which groups will be affected most by additional in-group data remained a challenge -- the challenge that this paper takes on. This paper presents an intuitive explanation that the degree to which group allocations will affect group-performance has to do with the relative separation of that group from other groups in the dataset. Formal results and several experiments back this claim up. It seems that the formal results only cover the case of two groups, but I'm guessing could be easily extended to more groups? Overall, this is an important problem and I appreciate that both a formal and experimental treatment is given. I have some concerns and questions over the presentation of the paper that I ask the authors to address during the rebuttal phase.

Notes:
- I have several points of concern and several questions. If this can be addressed during the rebuttal phase, I will consider changing my score.
- The appendix is very long. I did not have time to check it all.

**Strengths:**

1. This is an important problem and one that I do not think has been addressed yet, to my knowledge.
1. Including both the task-specific and the foundation model experiments adds depth to the experimental results and supports interesting comparisons.
2. I thought the limitations section was insightful and well thought out.
3. Formal statements are relevant and appear to be correct though I have not checked all details in the appendix.
4. Related work is sufficient to the best of my knowledge.

**Weaknesses:**

*1. Use of linear scaling laws.* The authors use linear fits to describe the relationship between subgroup allocation and group performance in figure 2 and figure 3. As the authors note, this is in deviation to past work (Rolf et al. 2021) -- additionally, this deviates from what one would expect from statistical learning theory. I understand that linear models are easier/less sensitive to fit. I disagree with the authors in their assessments that the trends in Figure 2 look linear. The linear fits seems to be particularly bad at the extremes, x-axis near 0 or 1, which are critical parts of the space for this type of analysis. In my opinion this is a major issue.

*2. Unclear what is being presented in figures 2/3*: The labels on figures 2 and 3 are inadequate. What does "subgroup balanced_acc mean"? I assume blue is group 1 and orange is group 2 in figure 2, but that needs to be states. Also, what does each group (1 or 2) correspond to in each dataset?

*3. Comparison to findings of past work in intro/conclusion:* I find statements like "Unlike standard explanations (e.g., assuming that under-represented or poorly performing-groups always benefit from increased data representation)..." a bit confusing. In my understanding, past work provides parametric forms for understanding how subgroup allocation affects subgroup performance. Those parameters could be such that adding more data from a subgroup does not affect sub-group accuracy, no? So why is this such a focus in comparing to past work.

**Questions:**

1. The authors mention that their analysis covers more than two groups (as does past work, like Rolf et al. 2021, FYI). But all experiments have two groupings. Could the authors please explain what they mean when they say " The multitude of subgroups we compare within the same dataset allows us to gain more insights than previous studies which usually only consider one or two standard groupings (Rolf et al., 2021; Claucich et al., 2025; Idrissi et al., 2022)."?
2. Why do you need to reduce the dimension of the embeddings with PCA (lines 330) before calculating TV? Why the 70% of the variance (that seems low)?
3. In interpreting the result in theorem 5.1, it's unclear to me how insightful this bound is to understanding model behavior. Is there some kind of trivial upper bound that doesn't require as many assumptions that the authors could compare to, and show that this bound is meaningful in comparison? (perhaps at the expense of additional assumptions?)
4. Could the authors comment on the assumtions part (ii) for theorem 5.1, are these reasonable in practice, and if so, why should we think so?

Suggestions:
- for Figures 5 and 6, it would be nice to report the actual TV for each condition, along with the arrow that shows TV is increasing left to right

---

> ### Author Response · Authors · 2025-11-18
> **Response to Reviewer Comment**
>
> We thank the reviewer for their thoughtful comments and the time they have put into reviewing this paper. We appreciate their positive comments on the importance of the problem we aim to tackle and the depth of our analyses and theory. We address each comment point-by-point below, and have also made revisions to our paper accordingly, which we have reuploaded.
>
> > Point 1: Extending formal results to more than two groups.
>
> We thank the reviewer for highlighting this, indeed it is relatively straightforward to extend our theory to settings with more than two discrete groups. In this case, the same assumptions hold, and instead of calculating the $TV(P(Z|Y,A=1), P(Z|Y,A=0))$, we determine the maximum TV over each pair in our set of groups A, and take this as our bound epsilon, with the rest of the theorem being unchanged. **We have included a formal statement on this in Appendix E2.3.**
>
> > W1: Use of linear scaling laws.
>
> We understand the reviewer’s concern with linear fits. We recognise they are imperfect, and as the reviewer notes, in some cases they are less accurate near the extremes. We tried more complex fits, including log-log, exponential, and the power law model used in Rolf et al. [1], however each time we obtained unstable fits. We expand on this in Appendix C2 and give an example in MIMIC where we fit Rolf’s model exactly as prescribed and obtain standard deviations of the order of 10,000, with parameters that are highly non-robust to small changes in how the fit is estimated or which data points are included (Table 5). We obtain similar problems for log-log and exponential fits. Our hypothesis is that due to the range of behaviours observed across subgroups, it is impossible to find one specific fit which works reliably. This is why we decide to maintain linear fits, as although they are sometimes imprecise, **they reliably and consistently capture the first-order pattern, telling us whether performance changes across allocations and how much it changes**. This is sufficient to capture a strong and consistent relationship to latent representation separation, and we leave more fine-grained analyses to future work. If the reviewer has other suggestions, we are happy to hear them. In any case, we have now expanded on this limitation in our Appendix C.
>
> > W2: Unclear presentation figures 2 and 3.
>
> We thank the reviewer for highlighting this, **we have now clarified the presentation of both figures**, adding a legend to Figure 2, adding more detail to the figure legends, and also removing the groups 0/1 in Figure 3 as actually in this case it is not necessary to consider a subgroup in relation to its pair. In Figure 2, each group (0 and 1) corresponds to one of the values of the attribute (e.g., for the attribute sex, group 0 = male and group 1=female). We list all of these in Table 2 in the appendix. Please let us know if you think either figure still needs clarifying.
>
> > W3: Comparison to findings of past work in intro/conclusion.
>
> We would first like to clarify our interpretation of past work. While it is true that Rolf et al. [1] provide parametric forms to show how subgroup allocation affects performance, as far as we know, they are the only ones who do so. Furthermore, while their work highlights that optimal subgroup allocations can vary, **they provide no explanation for this**. and require many models to be trained at different allocations to estimate a fit. **We have now clarified this distinction in our related work**. On the other hand, “standard” ML fairness often show examples of balancing subgroups (which sometimes work and sometimes do not) and describe reasons or hypotheses for this behaviour (without clearly proving them), such as that over-representing an under-represented group will improve its performance [2-6]. For instance, [5] says “It is obvious that increasing the proportion of T2 [under-represented group] will likely increase the performance on T2”. Therefore, by “standard explanations”, we are referring to the latter more common ML fairness papers which provide some form of explanation for subgroup performance. Upon re-reading the paper it seems like the use of e.g., “standard explanations” refers back relatively clearly to section 4 (“Current explanations are unreliable”), however please let us know if you think this should still be reworded.
>
> [1] Rolf et al., Representation Matters: Assessing the Importance of Subgroup Allocations in Training Data, ICML 2021.
>
> [2] Buolamwini et al., Gender Shades: Intersectional Accuracy Disparities in Commercial Gender Classification, PMLR 2018.
>
> [3] Idrissi et al., Simple data balancing achieves competitive worst-group-accuracy, PMLR 2022.
>
> [4] Kirichenko et al., Last Layer Re-Training is Sufficient for Robustness to Spurious Correlations, ICLR 2023.
>
> [5] Wang et al., Overwriting Pretrained Bias with Finetuning Data, ICCV 2023.
>
> [6] Alabdulmohsin et al., CLIP the Bias: How Useful is Balancing Data in Multimodal Learning?, ICLR 2024.

---

> > ### Author Response · Authors · 2025-11-18
> > **Response to Reviewer Comments (continued)**
> >
> > > Q1: Unclear "multitude of subgroups” when only use binary groups.
> >
> > We understand the confusion, apologies, we were referring to the fact that we consider multiple attributes (e.g., sex, age, dataset, ethnicity etc.) for a given dataset/task. However, as you point out, we only experimented with binary attributes (e.g., male/female, young/old, dataset 1/dataset 2, white/non-white). **We have reworded the line** you mention so it says “the multitude of attributes we compare within the same dataset …”, which should avoid this confusion. We wanted to highlight this because fairness papers often consider only one or two attributes for a given dataset/task/model. Here, **we compare many different attributes at once** which allows us to show that different subgroup allocation behaviours are specifically due to differences in the subgroup rather than just general differences in the model used or task considered.
> >
> > > Q2: Dimensionality reduction before calculating TV.
> >
> > This is a good point. We do not need to reduce the dimensionality of the embeddings with PCA before calculating TV. We choose to do this for two reasons, a) to reduce noise (in some cases, e.g, HAM, we have fewer test samples than the dimensionality of the embeddings), and  b), for computational efficiency (>10x speedup). However, **we checked that our estimates are robust to the dimensionality reduction process in two ways and added this to Appendix F**. First, we calculate the TV on the full feature space and find that it is highly correlated to the TV on the dimension reduced space (Spearman Rho > 0.99) (Figure 12). We also show how the TV changes as the number of principal components on which it’s calculated increases, and find that the absolute TV slightly decreases in some cases but the relative ordering across groups is stable (Figure 13).
> >
> > > Q3: How insightful is the bound in theorem 5.1?
> >
> > We thank the reviewer for highlighting this point. We first think the theorem is **insightful conceptually in connecting latent representation separation to subgroup performance change**. In terms of its practical usefulness, it is difficult to compare it to other upper bounds as this is a novel connection. The trivial upper bound would be that the accuracy difference is bounded by 1. We calculate the bound values in practice and find that for most subgroups we obtain meaningful bounds between 0 and 0.4 (**now included in Appendix E Figure 9**).
> >
> > > Q4: Are the theorem’s assumptions reasonable?
> >
> > The theorem’s assumptions are that fine-tuning datasets do not differ in P(Y) or P(Y|A). The assumption that P(Y|A) is stable should always be satisfied if samples from A are randomly selected when re-allocating the dataset. The assumption that P(Y) is unchanged is more restrictive in practice. It is satisfied if both groups A=0 and A=1 have the same distribution of Y (ie P(Y|A=0)=P(Y|A=1)). This is true for 10 out of 25 of the subgroups in our experiments (details are shown in Appendix B Table 2). While we relax this assumption for our experiments, and find the same consistent trend, we must make it for the purposes of the theorem, in particular so that we can isolate the effects of changes in A rather than changes in the label Y.
> >
> > > S1: Reporting actual TV in figures 5 and 6
> >
> > We thank the reviewer for this suggestion, and have edited figures 5 and 6 accordingly.
> >
> > We hope these manuscript revisions and clarifications give the reviewer confidence to raise their score and are happy to continue to engage further.

---

> ### Comment · Reviewer_5CBB · 2025-11-25
>
> Thank you for the responses! I have read through them and I don't have any further questions at this time.

---

### Official Review · Reviewer_ZvaL · 2025-11-01

**Soundness:** 2
**Presentation:** 4
**Contribution:** 2
**Rating:** 4
**Confidence:** 3

**Summary:**

The paper studies when balancing subgroup data affects model performance and proposes the latent separation hypothesis, suggesting that similar subgroup representations lead to less sensitivity to data allocation. It provides a theoretical bound under last-layer fine-tuning and tests the idea across several datasets and pretrained deep learning models. Results show consistent correlations between representation similarity and performance sensitivity, though within simplified and controlled setups.

**Strengths:**

The paper presents an attempt to explain when balancing subgroup data affects model performance. The main idea to link subgroup performance sensitivity to the similarity of their latent representations is clear. The experiments are consistent across datasets and the analysis is carefully done, though mostly within limited and controlled setups. The paper is clearly written and the results are easy to follow.

**Weaknesses:**

1. Doesn’t account for subgroup difficulty.
The paper assumes subgroup performance differences come mainly from representation separation, but it doesn’t consider that some subgroups might just be inherently harder to learn. For example, a subgroup with noisier labels or less distinctive features may naturally require more data. Without checking this, the correlation between latent separation and sensitivity could partly reflect difficulty, not just representation overlap.

2. Fixed fine-tuning budget is unrealistic
The experiments always use a fixed total fine-tuning size K, varying only how samples are distributed between subgroups. That makes sense for clean comparisons but doesn’t reflect real scenarios, where we often expand data rather than just reallocate it. The appendix includes a smaller K ablation and shows similar trends, which is nice, but this should be tested on more datasets or at larger budgets. A cost–benefit analysis across varying total sizes would make the setup more practical.

3. Theory limited to last-layer fine-tuning
The main theorem assumes only the classifier head is fine-tuned, keeping the encoder fixed. That’s quite narrow, since full-network fine-tuning is now standard. They do check this empirically and find similar correlations under full fine-tuning, but the theory doesn’t extend there. A relaxed version of the theorem that accounts for small feature drift, or an empirical measure of how stable subgroup embeddings remain during full fine-tuning, would strengthen the contribution.

4. Models are not true foundation models.
The main backbones (small CNN, DenseNet121, ViT-B/16, BERT-base) are standard pretrained networks, not foundation models. The only FM-style test is with CheXagent embeddings, where a linear head is trained on frozen features. While this shows the idea may generalize, it’s still far from real downstream FM use cases. The generalization claim should be softened, or an extra test with end-to-end fine-tuning on a true FM should be added.

**Questions:**

1. Can you measure or control for inherent subgroup difficulty (e.g. subgroup-level learning curves or probe accuracy) to ensure latent separation isn’t just reflecting hardness?
2. Why assume a fixed fine-tuning budget? Would the same correlation trends hold when the total dataset grows instead of being reallocated?
3. How much does the representation actually move under full-network fine-tuning? Can you quantify feature drift to connect theory and practice?
4. Could you extend the theory to full-network fine-tuning by introducing a small deviation term for representation change?
5. To back up the “foundation model” claim, can you try a second FM or an end-to-end low-rank fine-tuning on CheXagent or another FM?

---

> ### Author Response · Authors · 2025-11-18
> **Response to Reviewer Comment**
>
> We thank the reviewer for their feedback, which we respond to below with references to the revised manuscript we have re-uploaded. We hope our responses and additional experiments will give the reviewer confidence to raise their score.
>
> > W1/Q1: Does not account for subgroup difficulty.
>
> We completely agree with the reviewer that subgroup difficulty should be accounted for, and that more difficult (e.g., noisier) subgroups may require more data to be learnt well. We investigated this hypothesis directly, and we kindly **refer the reviewer to Figure 3 (bottom panel)**, which we discuss in Section 4.3. This figure shows the initial model’s performance on each subgroup (proxy for difficulty) compared to how the subgroup’s loss changes as its fine-tuning data allocation is increased (e.g., very negative loss slope means big improvement in performance as data allocation is increased). We show that **subgroup difficulty is an inconsistent predictor of sensitivity to allocation**. For example, while one low-performing subgroup in Civil_comments has a loss which decreases substantially at higher allocations, other difficult subgroups show no change in allocation. This very inconsistency is what motivated us to explore alternative explanations.
>
> >W2/Q2: Why assume a fixed fine-tuning budget?
>
> We thank the reviewer for this comment, and we agree that studying how subgroup performance changes in settings where data is expanded would be an interesting, complementary research question (we have now noted this in our limitations and future work section). However, for this work, we choose to focus on the fixed-budget setting for two reasons. First, it allows us to specifically **isolate the effect of re-balancing subgroup data**, providing a clean, fundamental analysis of why these effects can differ. Secondly, this setting is **highly practical as many real-world use cases rely on subgroup re-balancing at a fixed budget**, including standard bias mitigation methods like resampling or groupDRO and their derivatives [1,2,3], and settings where one needs to decide what future data to collect, curate, or synthesise [4,5]. Moreover, we conducted ablations on MIMIC (our largest dataset with the most subgroups) where we repeated our experiments with a smaller fine-tuning budget and found equally strong trends (Appendix J).
>
> > W3/Q3: How much does the representation actually move under full network fine-tuning?
>
> We thank the reviewer for this interesting question. We explored how the representations change under full network fine-tuning at different allocations and found that surprisingly, the **representations were very stable**. In Appendix H **Figure 15** we show how the TV distance across attributes changes for different fine-tuning allocations, and find that there are very little shifts, with the distance remarkably stable even at the allocation extremes. We believe this explains why we still observe very high correlations with latent representation separation in the full fine-tuning setup.
>
> > W3/Q4: Could you extend the theory to full network fine-tuning?
>
> This is another interesting point. **The theory could indeed be extended to an arbitrary collection of last layers (including the full network)** however this would mean the representations Z on which TV is computed would have to shift accordingly (e.g. would be calculated on X in the case of full network fine-tuning). This may not necessarily yield meaningful bounds particularly if it is computed on the input space. We have added this remark to the theory section of our revised paper.
>
> To extend the theory to full network fine-tuning while considering TV on a latent space Z is non-trivial, and a valuable future work direction. That being said, as we noted in our previous answer, our empirical results support this line of reasoning, most likely due to the fact that there is very little feature drift.
>
> [1] Idrissi et al, Simple data balancing achieves competitive worst-group-accuracy, PMLR 2022.
>
> [2] Kamiran et al, Data preprocessing techniques for classification without discrimination, Knowledge and Information Systems 2012.
>
> [3] Kirichenko et al, Last layer re-training is sufficient for robustness to spurious correlations, ICLR 2023.
>
> [4] Ktena et al, Generative models improve fairness of medical classifiers under distribution shifts, Nature medicine 2024.
>
> [5] Wang et al, Overwriting Pretrained Bias with Finetuning Data, ICCV 2023.

---

> > ### Author Response · Authors · 2025-11-18
> > **Response to Reviewer Comment (continued)**
> >
> > > W4/Q5: To back up the “foundation model” claim, can you try a second FM or an end-to-end low-rank fine-tuning on CheXagent or another FM?
> >
> > We thank the reviewer for this constructive suggestion. We **have now started experiments on a second FM**, and will get back to you with results when they are ready. We agree that our main contribution is on standard single task models, and are happy to soften any claim on FM generalisation. Currently we do not see any particularly overstated claims; our 4th contribution is simply that our findings “can guide dataset selection decisions [...] in a practical case-study fine tuning a VLM”, but please point us to any that you think we should modify.
> >
> > Please let us know if this has resolved your questions, we are very happy to discuss further.

---

> > > ### Author Response · Authors · 2025-11-25
> > > **Response to Reviewer Comment (follow-up)**
> > >
> > > As the reviewer suggested (W4/Q5), we conducted experiments on a second foundation model to confirm the generalisability of our hypothesis across both single-task models and larger vision-language models. We implemented Microsoft's RAD-DINO-MAIRA-2 image encoder [1] to extract MIMIC embeddings and train a classification probe on top of them.
> > >
> > > We found **very similar trends to the previous foundation model, where increased latent TV distance is significantly associated with increased sensitivity to subgroup allocation**. For the most-separated groups (i.e. imaging-related), selecting. a balanced allocation can improve subgroup accuracy by up to 0.03 points, while for, e.g, gender, the accuracy gain is less than 0.01. **We have added these results to Appendix I (Figures 21, 22, and 24) and referred to them in the main text.**
> > >
> > > Overall, both of our results in foundation model fine-tuning provide strong evidence for our hypothesis in a more extreme setting where the pre-training and fine-tuning datasets are drawn from different distributions, and show how our hypothesis can help inform fine-tuning dataset selection or balancing in practice.
> > >
> > > We thank the reviewer for this suggestion and think it has strengthened our manuscript. We hope that these results, along with our responses and results mentioned above, will give the reviewer confidence to raise their score, and remain open to any further discussion.
> > >
> > > [1] Perez-Garcia et al., Exploring scalable medical image encoders beyond text supervision, arXiv 2024.

---

### Official Review · Reviewer_Pxvq · 2025-11-04

**Soundness:** 3
**Presentation:** 2
**Contribution:** 2
**Rating:** 4
**Confidence:** 3

**Summary:**

The authors study the problem of subgroup performance under group imbalance. Specifically, they theoretically investigate the conditions on which balancing the dataset by the subgroup variable improves the accuracy of each subgroup. The critical quantity is the TV distance between the latent distribution across groups for each class, where lower separability means reduced benefit from data balancing. The authors empirically validate their theory on four datasets.

**Strengths:**

1. The paper tackles an important problem in algorithmic fairness.

2. The authors derive a simple and intuitive TV upper bound.

**Weaknesses:**

1. Though Theorem 5.1 provides some good intuition, it does not sufficiently characterize the empirical phenomenon studied in the paper. In particular, the authors should derive a bound on the slope of the per-subgroup balanced acc vs. subgroup proportion (Figure 2), as a function of the TV or some other terms.

2. The assumption that $P(Y)$ stays the same across the two datasets in L290 seems very strong. In particular, if we have $P(Y | A= 0) \neq P(Y | A= 1)$, it seems like changing the subgroup allocations will necessarily change $P(Y)$.

3. Prior works in spurious correlations have characterized the phenomenon where mitigating one bias will exacerbate another. The authors should characterize their theorem in this setting, e.g. when does balancing wrt one attribute increase the gap in another attribute?

4. The authors state in the remark that $|\Delta Acc|$ is small empirically relative to the subgroup performance change. I am not able to find the empirical validation of this claim.

5. The authors should further motivate the practical utility of the method. It seems like the primary setup is when a practitioner has a large set of possible attributes to balance, and they can compute the TV first and use it as a filter, i.e. only try balancing attributes with high TV.  However, this seems unlikely since we normally know in advance which attributes we must be fair to. In addition, if it is just retraining the last layer, the compute time of calculating TV may not be much less than just finetuning it anyways.

[1] Li, Zhiheng, et al. "A whac-a-mole dilemma: Shortcuts come in multiples where mitigating one amplifies others." Proceedings of the IEEE/CVF Conference on Computer Vision and Pattern Recognition. 2023.

**Questions:**

1. Please provide more details on how the TV is computed in practice using the PCA approach. Is there a way to quantify the estimation error resulting from this?

2. How important is the last-layer retraining assumption here? If I replaced $Z$ with $X$ in Theorem 5.1, and had a way to compute TV in high dimensions, could I then choose $g_{\theta}$ to be my whole network?

---

> ### Author Response · Authors · 2025-11-18
> **Response to Reviewer Comments**
>
> We greatly appreciate the reviewer’s thoughtful comments. We have incorporated these suggestions into our revised manuscript, which we re-uploaded, and responded to each point below.
>
> >W1: The theorem 5.1 does not sufficiently characterize the empirical phenomenon studied in the paper.
>
> We thank the reviewer for their suggestion. We agree that the theorem does not characterise the full empirical phenomenon, however **deriving a bound on the slope would require extra assumptions on training dynamics that are not justified and would narrow the theorem’s scope**. The theorem’s purpose was to show when subgroup allocation should not matter and give a theoretical intuition for the results we observe, and we believe it is sufficiently informative for its purpose.
>
> >W2: The assumption that P(Y) stays the same across the two datasets in L290 seems very strong.
>
> The reviewer is correct that our assumption is invalidated if $P(Y|A=0) \neq P(Y|A=1)$. We impose this assumption in our theory so that we can isolate the effects of changes in A rather than changes in the label $Y$. We note that in practice, there are real subgroups for which $P(Y|A=0) = P(Y|A=1)$. For example in our experiments, 10 out of 25 of the subgroups approximately satisfy this. **We have clarified this in Table 2 of our appendix to specifically highlight which subgroups do or do not satisfy this assumption.** Of course, many subgroups will not satisfy this assumption. Our broader experiments include these cases, and we still observe the same relationship.
>
> >W3: The authors should characterize their theorem in this setting, e.g. when does balancing wrt one attribute increase the gap in another attribute?
>
> We thank the reviewer for highlighting this interesting point. According to our theorem, balancing w.r.t. one attribute B affects performance on attribute A if this changes the effective allocation of A (P(A)). If balancing B alters P(A) (e.g., when A and B are correlated), then the same reasoning in Theorem 5.1 applies, and the differences in subgroup accuracies across A will be bounded by the representation separation for A. Conversely, if B and A are independent, balancing B should not affect performance on A. **We have added this interpretation and citation to our discussion of the theorem in Appendix E2.1.**
>
> >W4: Empirical validation of the claim that delta ACC is small relative to subgroup performance change.
>
> We thank the reviewer for noting this, this was an oversight on our part, **we have now added this analysis to the Appendix E2.3 (Figure 8) and referred to it in the theory section**. We see that indeed there is very little variation in overall model performance across fine-tuning allocations, with a mean accuracy standard deviation of 0.016, 0.002, 0.001, and 0.001 for MNIST, MIMIC, HAM, and Civil_comments respectively. We point this out as evidence to suggest that the upper bound in subgroup accuracy difference at different fine-tuning allocations is primarily driven by $4\epsilon$, i.e., TV differences.
>
> >W5: The authors should further motivate the practical utility of the method…
>
> The primary practical utility of our method is **to help inform future data collection, curation, balancing or synthesis decisions, to improve a base model**. If a practitioner has trained a model and observes that the model is not fair with respect to certain attributes they care about (e.g., sex and ethnicity), they may decide to try to improve its performance by fine-tuning it on more data. If they observe for instance that the model has very low TV distance for sex but not ethnicity, they could decide to allocate resources to collecting balanced data with respect to sex but not necessarily ethnicity. Similarly, if they found low TV distance for both attributes, they could simply allocate efforts to collecting high quality data for fine-tuning.
>
> The reviewer mentions that “we normally know in advance which attributes we must be fair to”; however we do not see how this takes away from the practical value of our work. Whether one cares about being fair to 1 or 10 attributes, our work would still be practically useful in deciding which subgroups to prioritise, if any.
>
> With respect to computational cost, if the data has not been collected, curated, or synthesised the cost gains are evident. If the data has already been collected and the question is just about balancing it (e.g., as in [1,2]), it can still be efficient in informing what type of balancing to do instead of fine-tuning multiple models and using the best one (**TV calculation is approximately 10- to 20-fold faster than last-layer fine-tuning of a single model**).
>
> **We agree that this is an important point to clarify, and we have expanded upon this in section 6.4 of the revised paper.**
>
> [1] Kirichenko et al, Last Layer Re-Training is Sufficient for Robustness to Spurious Correlations, ICLR 2023.
>
> [2] Wang et al., Overwriting Pretrained Bias with Finetuning Data, ICCV 2023

---

> ### Author Response · Authors · 2025-11-18
> **Response to Reviewer Comments (continued)**
>
> > Q1: How is TV computed and is there a way to quantify the estimation error?
>
> We compute TV by first extracting $n \times$ *last_layer_dim* (e.g., 1024 for DenseNet) dimensional latent vectors $z$ for each of our test samples. We use PCA to reduce the dimensionality of this to the top-K components which encode 70% of the variance. This gives us $n$ lower dimensional vectors of size 2 to 81 (depending on the model and data). For each component vector, we calculate the TV distance between samples where $A=0$ and $A=1$, conditioning on $Y4, and average this across all components. A full explanation with the relevant equations is presented in Appendix F1. We will also release our code upon acceptance with full implementation details.
>
> With respect to estimation error, is the reviewer referring to estimation error from using the top PCs instead of the full high dimensional embeddings? If this is the case, yes we did explore this, and found that our results were robust to our dimensionality reduction process. We have now added extra graphs to Appendix F3 showing that a) **the TV calculated on top K principal components is highly correlated to the TV calculated on the full feature space (spearman rho > 0.99, Figure 12)** and that b), if the TV is calculated on more principal components (eg those which explain 80 - 100% of the variance), while the absolute TV slightly decreases in some cases, **the relative ordering across groups is stable (Figure 13)**. We do not believe that this introduces an estimation error, if anything it reduces variance in the empirical TV estimate (particularly given our limited test dataset size) and reduces noise.
>
> > Q2: Importance of last-layer re-training assumption.
>
> This is a good point, the last-layer re-training assumption is not important. **The theorem could trivially be extended to an arbitrary collection of last layers** assuming TV can be computed on the appropriate embedding space (or X if it was the full network). We use the last-layer as this is often done in fine-tuning and has been discussed in the bias mitigation literature [1].
>
> However, doing it on the full network or its early layers would most likely not make sense empirically as the TV estimate is likely to be large regardless, and would not necessarily reflect the representations the model actually uses for the task. For instance, for the MNIST parity classification task, the TV between red and green images is large in the input space but not in the embedding space, so the upper bound would not be informative if the full network was used.
> We thank the reviewer for highlighting this point, we have added a remark to the theory section of the paper.
>
>
> We hope these explanations and paper edits will give the reviewer confidence to raise their score, and in any case we are very happy to further engage if there are other points of clarification or concern.
>
> [1] Last Layer Re-Training is Sufficient for Robustness to Spurious Correlations, ICLR 2023.

---

> > ### Author Response · Authors · 2025-11-27
> > **Follow up**
> >
> > We thank the reviewer again for their thoughtful initial review. We believe the manuscript revisions and clarifications posted on Nov 18th, including the extra additions to the theory, the reiteration of the our method’s practical utility, and the further experiments on TV, have strengthened the paper and addressed their comments. We would appreciate if they could confirm that their concerns have been resolved or whether they have any further feedback.

---

### Author Response · Authors · 2025-12-03
**Summary of Responses and Revisions**

Dear AC,

We summarise the steps we have taken to address the five reviewers’ comments below. We are grateful for their feedback, which has **substantially strengthened the revised manuscript** (changes marked in blue). While the shortened rebuttal period limited discussion, we are encouraged that **Reviewer zYwC raised their score**, and we are confident that the revisions listed below also address the other reviewers’ concerns.

All five reviewers received the paper positively, highlighting that we “tackle an **important problem**” (Reviewers Pxvq, 5CBB, zhu1), with an “**intuitive**” hypothesis (Reviewers Pxvq, 5CBB, zYwC), which is “**well justified both theoretically and empirically**” (Reviewers 5CBB, zhu1, zYwC), displaying a commitment to “**scientific rigor**” (Reviewers zhu1,zYwC).

Overall, our revisions have significantly **increased the scope of our findings**. Theoretically, we broadened our results to include cross-attribute effects, non-binary groups, and partial/full fine-tuning. Empirically, we further validated our hypothesis when training-from-scratch and on a second foundation model, showed new results on representation stability after full fine-tuning, and demonstrated the causal nature of our hypothesis by directly intervening on the TV distance.

**1. Reviewer Pxvq (initial score: 4, no further engagement before leak)**
* **Theoretical remarks**: we explicitly detailed our **assumption’s validity** (Table 2 and Sec. E2.1), showed empirical validation that $|\Delta \text{Acc}|$ was small (Fig. E9), and added remarks on **cross-attribute effects** (Sec. E2.2) and **extensions to an arbitrary set of last layers** (Sec. 5).
* **Robustness of TV calculation**: we demonstrated that the TV calculated on the top-K principal components and on the full feature space **correlate highly** (Spearman's $\rho > 0.99$, Fig. F13).
* **Practical utility**: we reiterated how our method can inform data collection, curation, or balancing decisions, with limited computational overhead (Sec. 6.5).

**2. Reviewer ZvaL (initial score: 4, no further engagement before leak)**
* **Additional experiments**: we validated our hypothesis on a **second foundation model** (Microsoft’s RAD-DINO-MAIRA-2) which showed equally strong results (Sec. I.2) and added further analyses on the **stability of representation distances after full fine-tuning** (Fig. H16).
* **Clarifications**: we clarified we had accounted for subgroup difficulty (Fig. 3), justified the use of a fixed fine-tuning budget, and extended the theory to full network fine-tuning (remark in Sec. 5).

**3. Reviewer 5CBB (initial score: 4, acknowledged responses with no further questions)**
* **Theory extensions**: we included a formal statement on **extending results to > 2 groups**, (Sec. E2.3), showed our **bound was insightful** in practice (Fig. E9), and included a discussion on our assumptions (Sec. E2.1).
* **Use of linear scaling laws**: we further justified this in Sec. C2, detailing why **other fits were not as robust** as ours.
* **Clarifications**: we clarified our distance calculation method and showed it is highly robust (Figs. F13 and F14), and expanded on how our work differs from Rolf et al., who provide no explanation for why sensitivity to allocation varies.
* **Presentation**: we improved the presentation of Figs. 2 and 3 and reported actual TV values in Figs. 5 and 7.

**4. Reviewer zhu1 (initial score: 4, no further engagement before leak)**
* **Scope extension**: we confirmed that our **hypothesis also holds in the training-from scratch** setting (Figs. H19 and H20) with effects of an even stronger magnitude across our four datasets.
* **Evaluation of representations**: we showed **representation distances are remarkably consistent** after fine-tuning (Fig. H16) and elaborated on this in Sec. H1.

**5. Reviewer zYwC (initial score 4, raised to 6 on 26 Nov)**
* **Operationalisation of hypothesis**: we included a new experiment showing that explicitly **enforcing low TV leads to lower sensitivity** to allocation, suggesting our findings go beyond correlation (Fig. 6).
* **Scope of findings**: we clarified why our hypothesis was enough to answer key practical questions (Sec. 5.7) and why studying mechanisms which affect representation separation, while related, is not the focus of this work (Sec. 5.7).
* **Contextualisation**: we elaborated on how our work shares the premise of robust representation learning methods while exploring a different angle (Sec. 6.3) and incorporated the suggested references.
* **Clarifications**: we implemented the suggested changes, including **improving our theory’s notation**.

**Conclusion**: Our paper provides a novel and intuitive hypothesis which addresses the important and unsolved problem of understanding why subgroup balancing has variable effects. We thank the AC for their consideration.

---

### Meta-Review · Area_Chair_Q7oc · 2026-01-03

**Summary:**

In their initial review, all reviewers were leaning negative. However, there was significant author rebuttal to address concerns.

Reviewers broadly agree that this is an important problem, that the idea is simple and intuitive, and that the paper is well-written; also some comment on how well / thoroughly the experiments are conducted.

Many smaller concerns were well-rebutted by the authors. Of the more major ones, a few crucial points remain in my opinion, relating to the somewhat limited scope of the paper (pretrain-then-finetune setting):
1. The theory is straightforward / unsurprising, with little/no key insights afforded, and no obvious ability to extend it (eg beyond last layer);
2. Subgroup difficulty should matter (reviewer ZvaL), and I do not think the authors rebutted this point sufficiently (eg if one subgroup has noisier labels, then how does that link to needing more data and to latent representation);
3. Assumption may not be reasonable (P(Y|A) constant) but this is a minor point, as the authors did provide further analysis;
4. A few times the authors said this was out-of-scope or not in other papers (eg why does subgroup performance sometimes increase). Given the relatively weak impact of the theory, I feel like addressing one or more of these questions in the paper would make it stronger. Another example is looking into why some subgroups overlap naturally while others do not?

Authors did rebut other points well, including adding another experiment in response to reviewer zYwC (although taking this further would improve the paper more). Other examples include clarifying some figures, notation and experimental details/choices; experiment with something more like a foundation model; clarifying use-case clearly.

**Reviewer Concerns:**

Please see summary review for a list of concerns addressed and not sufficiently addressed in my opinion.

**Reviewer Scores:**

Pxvq: stay at 4 as 2 main concerns were not sufficiently rebutted.

ZvaL: Stay at 4.

5CBB: Either stay at 4 or increase to 6.

zhu1: Stay at 4 (limited scope).

zYwC: Probably increase to 6 (reviewer states that they increased score in a comment); additional experiment by authors likely caused this.

---

### Decision · Program_Chairs · 2026-01-26

Reject